



# An all-sky 1 km daily surface air temperature product over mainland China for 2003–2019 from MODIS and ancillary data

Yan Chen[1], Shunlin Liang[2], Han Ma[1], Bing Li[1], Tao He[1], Qian Wang[3,4]

[1]School of Remote Sensing and Information Engineering, Wuhan University, Wuhan 430079, China
[2]Department of Geographical Sciences, University of Maryland, College Park, MD 20742, USA
[3]State Key Laboratory of Remote Sensing Science, Beijing Normal University, Beijing 100875, China,
[4]Institute of Remote Sensing and Digital Earth, Chinese Academy of Sciences, Beijing 100875, China

*Correspondence to*: Shunlin Liang (sliang@umd.edu)

**Abstract.** Surface air temperature ($T_a$), as an important climate variable, has been used in a wide range of fields such as ecology,
hydrology, climatology, epidemiology, and environmental science. However, ground measurements are limited by poor spatial representation and inconsistency, while reanalysis and meteorological forcing datasets suffer from coarse spatial resolution and inaccuracy. Previous studies using satellite data have mainly estimated $T_a$ under clear-sky conditions, or with limited temporal and spatial coverage. In this study, an all-sky daily mean $T_a$ product at 1 km spatial resolution over mainland China for 2003–2019 has been generated mainly from the Moderate Resolution Imaging Spectroradiometer (MODIS) products and
the Global Land Data Assimilation System (GLDAS) dataset. Three $T_a$ estimation models based on random forest were trained using ground measurements from 2384 stations for three different clear-sky and cloudy-sky conditions. The validation results showed that $R^2$ and root mean square error (RMSE) values of the three models ranged from 0.984 to 0.986 and 1.342 K to 1.440 K, respectively. We examined the spatiotemporal patterns and land cover type dependences of model accuracy. The relative contributions of different features to models were also quantitatively analysed. Finally, values of our $T_a$ product in
2010 were validated and compared with the China Land Data Assimilation System (CLDAS) dataset at 0.0625° spatial resolution, China Meteorological Forcing Data (CMFD) dataset at 0.1° spatial resolution and GLDAS dataset at 0.25° spatial resolution. The $R^2$ and RMSE values of our product were 0.992 and 1.010 K, respectively, indicating this high-resolution satellite product has significantly higher accuracy. In summary, the all-sky daily mean $T_a$ dataset developed in this study has achieved satisfactory accuracy and high spatial resolution simultaneously, which fills the current dataset gap in this field and
plays an important role in the studies of climate change and hydrological cycle. This dataset is freely available at http://doi.org/10.5281/zenodo.4399453 (Chen et al., 2021b) and the University of Maryland (http://glass.umd.edu/Ta_China/) currently. A sub-dataset that covers Beijing generated from this dataset is publicly available at http://doi.org/10.5281/zenodo.4405123 (Chen et al., 2021a).




## 1 Introduction

Surface air temperature ($T_a$) is one of the most important variables in a wide range of fields including ecology, hydrology, climatology, epidemiology, and environmental science (Goetz et al., 2000; Stisen et al., 2007; Vancutsem et al., 2010; Zhang et al., 2018). $T_a$ refers to the atmospheric temperature 1.5–2 m above the surface, which represents the thermal state information of the surface and the lower atmosphere. It influences the carbon cycle through the biophysical effects of vegetation and regulates many surface processes such as photosynthesis, respiration, and evaporation (Khesali and Mobasheri, 2020). Reliable

estimates of $T_a$ at fine spatiotemporal resolution are importance to better understand and simulate complex surface processes and reveal changes due to climate change or local disturbances (Guan et al., 2013). Moreover, in the context of continuous global warming, meteorological disasters caused by frequent extreme weather events and consequential social and economic losses are increasing gradually. A deep understanding of the spatiotemporal patterns of $T_a$ is also of great guiding significance for disaster prevention and reduction.

However, because of its proximity to the interface between land/ocean and atmosphere, the near-surface air is influenced by various exchange processes between these three Earth system compartments (Schwingshackl et al., 2018). The spatiotemporal patterns of $T_a$ can vary and be complicated due to the heterogeneity of various environmental factors (such as solar radiation, latitude, underlying surface, cloud cover, and season) that impact the energy balance of the land–atmosphere system (Benali et al., 2012; Chen et al., 2015; Prihodko and Goward, 1997).

The $T_a$ data is one of the most frequent observation data recorded by meteorological stations. In situ $T_a$ usually has reliable accuracy and high temporal resolution; however, it has some flaws, such as limited spatial representation, measurement inconsistency, and uneven spatial distribution of ground stations (Jang et al., 2014; Prihodko and Goward, 1997). Geographical interpolation methods such as inverse distance weighting (IDW), kriging, and spline function have been widely used to estimate the spatial distribution of $T_a$ (Benavides et al., 2007; Ishida and Kawashima, 1993; Kurtzman and Kadmon, 1999). However,

these methods usually consider only the autocorrelation of $T_a$, ignoring the complex factors that lead to its heterogeneity. The accuracy of interpolated $T_a$ is greatly affected by station network density, which leads to relatively poor accuracy being obtained in areas with sparse stations (Stisen et al., 2007; Vogt et al., 1997). Therefore, the accuracy of interpolated $T_a$ may have significant errors associated with unrepresentative spatial patterns, and there can be great uncertainty in describing the spatial patterns of $T_a$ over large areas in this way (Benali et al., 2012; Rao et al., 2018).

Remotely sensed data have provided unprecedented spatial coverage at regional and global spatial scales (Liang, 2004). Over the past few decades, many schemes have been developed to estimate $T_a$ from remotely sensed data. The strong physical relationship between the land surface temperature (LST) and $T_a$ has become the research basis of many $T_a$ estimation methods. Generally speaking, the LST-based $T_a$ estimation methods can be divided into three distinct categories. The first type is the traditional statistical method, including univariate regression method to establish a linear relationship between $T_a$ and LST,

and multiple regression methods considering various variables (such as solar zenith angle, elevation, Julian day, etc.) in addition to LST (Lin et al., 2012; Zeng et al., 2015). The second is the temperature-vegetation index (TVX) method, based on





the negative correlation between normalized difference vegetation index (NDVI) and LST in the study area (Stisen et al., 2007; Vancutsem et al., 2010; Zhu et al., 2013). The third is the land surface energy-balance physical method, which uses crop water stress index (CWSI) and the aerodynamic resistance to estimate $T_a$. This method has a good physical basis, but usually relies

on numerous input parameters (such as roughness, soil physical properties), which are always difficult to obtain (Sun et al., 2004). In principle, the atmospheric profile products from satellite observations include temperature profile of the entire atmosphere, but usually require additional processes to obtain $T_a$. The Moderate Resolution Imaging Spectroradiometer (MODIS) atmospheric profile product has been used for this purpose (Bisht and Bras, 2010; Borbas and Menzel, 2017; Zhu et al., 2017). Generally, traditional statistical methods were commonly used but have reported low accuracy. In recent years,

machine learning methods, particularly deep learning methods, such as support vector machine (Zhang et al., 2016), artificial neural network (Jang et al., 2004; Zhang et al., 2016), M5 model trees (Emamifar et al., 2013), random forest (RF) models (Noi et al., 2017; Xu et al., 2014; Zhang et al., 2016), cubist models (Meyer et al., 2016; Noi et al., 2017; Rao et al., 2019), and advanced deep learning methods (Shen et al., 2020), have been gradually applied to $T_a$ estimation from satellite data because of their stronger learning ability to capture the complex nonlinear relationship between various factors.

75       Most LST-based $T_a$ estimation methods mentioned above are suitable only for clear-sky conditions as the current LST datasets are mainly derived from satellite thermal infrared radiances (TIR) that are susceptible to cloud contamination (Liang et al., 2019; Ma et al., 2020). Currently, there are two main strategies for estimating all-sky $T_a$ based on LST: one is to derive $T_a$ from the available LST first and then fill the $T_a$ gaps (Rosenfeld et al., 2017; Zhang, 2017); the other is to first fill the LST gaps to develop a seamless product and then estimate the all-sky $T_a$ (Kilibarda et al., 2014; Li et al., 2018; Rao et al., 2019).

For example, Zhang et al. (2017) estimated $T_a$ under clear-sky conditions based on MODIS LST, and the Atmospheric Infrared Sounder (AIRS) standard $T_a$ products were used to fill the cloudy-sky pixels after a downscaling process, with a mean absolute error (MAE) of 1.2 K and root mean square error (RMSE) of 1.6 K overall. According to the research conducted by Kilibarda et al. (2014), the 8-day composite LST was interpolating into a daily dataset and then combined with topographic layers and geometric temperature trend to interpolate the all-sky daily $T_a$, and the results reported an RMSE value of 2–4 °C. In addition,

Zhu et al. (2017) developed a parameterization scheme to estimate all-sky instantaneous daytime $T_a$ only relying on MODIS atmospheric profile product. They first established the relationship between LST and $T_a$ under clear-sky conditions, and then estimated $T_a$ under cloudy-sky conditions based on the established relationship, with RMSE values ranging from 2.50 °C to 2.56 °C.

      Currently, several studies have been conducted to develop all-sky $T_a$ datasets based on remotely sensed data. For instance,

Li et al. (2018) used a 3-step hybrid gap-filling method to attain seamless LST first, and then developed daily geographically weighted regression (GWR) models to interpolate $T_a$ using gap-filled LST and elevation, and finally developed a 1 km daily minimum/maximum $T_a$ dataset in urban and surrounding areas in the conterminous U.S. for 2003–2016. The cross-validation results reported that the RMSE values were 2.1 °C and 1.9 °C for daily minimum and maximum $T_a$, respectively. In the recent work conducted by Yao et al. (2020), the MODIS 8-day composite LST was averaged to obtain monthly mean LST, and then

combined with enhanced vegetation index (EVI), solar radiation, topographic index and other features to establish a cubist

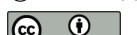



model for generating 1 km monthly maximum/mean/minimum $T_a$ products in China, and the RMSE of the estimated monthly mean $T_a$ was 0.629 °C. In addition, Rao et al. (2019) first filled the gaps of LSTs, and then used the gap-filled LSTs and some radiation products to build cubist models for estimating all-sky daily mean $T_a$, with an RMSE of 1.87 °C. Finally, a 0.05° × 0.05° daily mean $T_a$ product over the Tibetan Plateau for 2002–2016 was developed. In addition, there exists multiple

reanalysis and meteorological forcing datasets covering large areas or global areas, which are usually generated by data assimilation or data interpolation, such as the Global Land Data Assimilation System (GLDAS) (Rodell et al., 2004), Modern-Era Retrospective Analysis and Research and Application, version 2 (MERRA-2) (Gelaro et al., 2017), China Meteorological Forcing Data (CMFD) (Yang and He, 2019), and China Land Data Assimilation System (CLDAS) (Shi et al., 2011). However, these datasets have coarse spatial resolution (generally ≥ 0.1° except for CLDAS with a spatial resolution of 0.0625°) and

regional inaccuracy, which may limit their potential to accurately capture the spatial heterogeneity of $T_a$ in the urban and mountainous areas and lead to uncertainties for applications at local to regional scales (Jang et al., 2014; Li et al., 2018; Zhu et al., 2017). To our knowledge, there are a lack of long time series all-sky $T_a$ products covering vast areas with both high spatial and temporal resolution currently.

The main objective of this study is to develop an all-sky 1 km daily mean $T_a$ over mainland China for 2003–2019 by

integrating satellite data products, model simulations, and ground measurements. For the first time, assimilated $T_a$ was applied to supplement and substitute MODIS LSTs and provide the initial values of model prediction. In order to solve the issue of missing LST, a simple temporary filling method was used to fill the gaps of MODIS LSTs first. Considering the differences in the relationship between $T_a$ and other features under different weather conditions, we divided all data pairs into three types of weather conditions: (1) clear-sky conditions; (2) cloudy-sky conditions case I; (3) cloudy-sky conditions case II, and then

established three machine learning models to estimate daily mean $T_a$ under different weather conditions. The structure of this paper is organized as follows: Section 2 describes the study area and used data; Section 3 summarizes the overall research method; Section 4 reports the validation results and discusses the model performance; Section 5 compares the developed product with the existing datasets; and Sect. 6 presents the overall conclusion.

## 2 Data

### 2.1 Meteorological station data

This study was conducted in mainland China. Station observed daily mean $T_a$ from 2003 to 2019 were collected from 2384 standard meteorological stations in mainland China for model training and validation. During the production process of this dataset, it experienced strict quality control. Figure 1 shows the study area and the geographical locations of the meteorological stations used in this study.



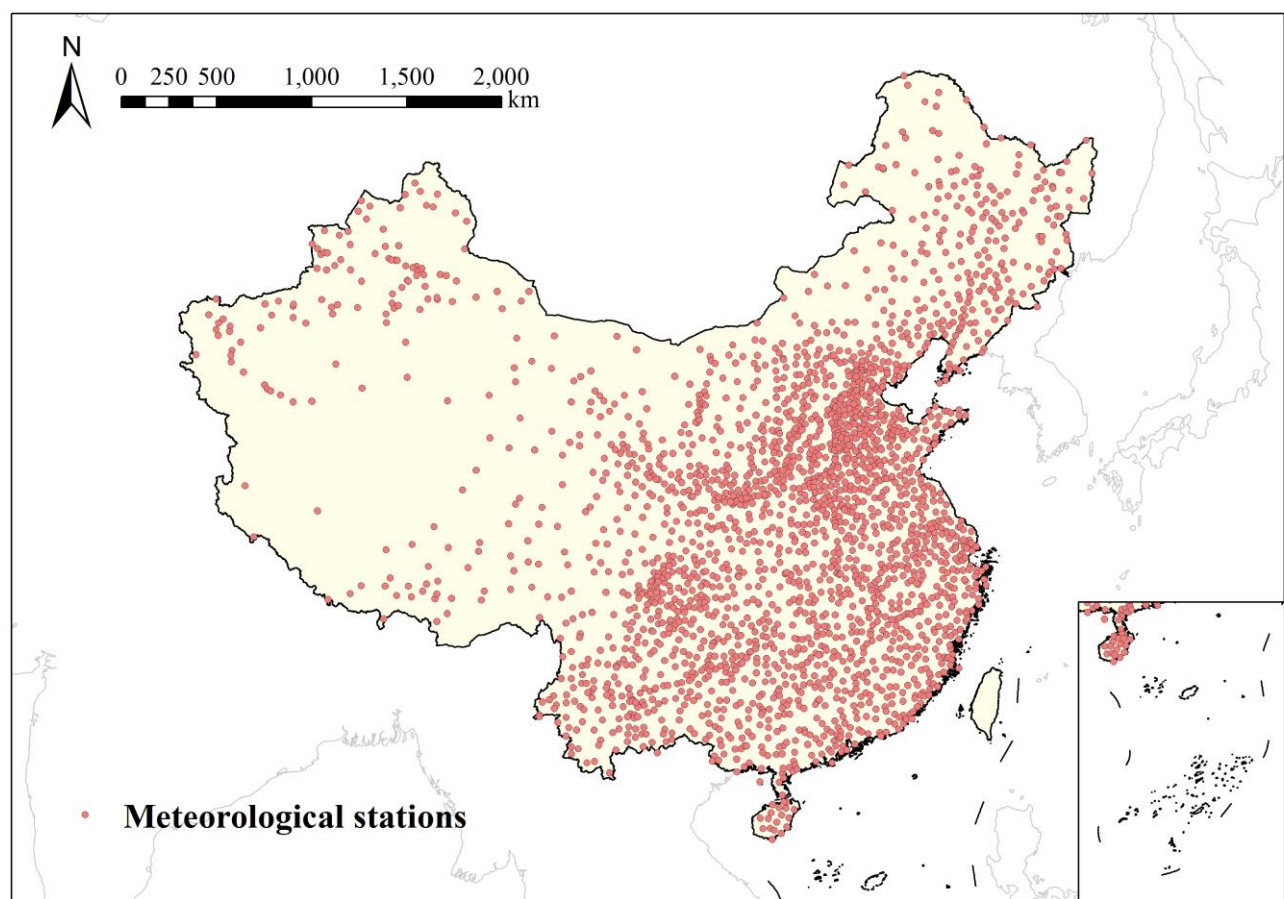


**Figure 1. Study area and meteorological station locations used in this study.**

## 2.2 Remotely sensed data

Satellite datasets used in this study are listed in Table 1.

**Table 1. Satellite datasets used in this study.**

| Product | Dataset(s) | Spatial resolution | Temporal resolution |
|---|---|---|---|
| Land surface temperature (LST) | MOD11A1, MYD11A1 | 1 km | Daily |
| Downward shortwave radiation (DSR) | GLASS05B01 | 0.05° | Daily |
| Surface albedo (ALB) | GLASS02A06 | 1 km | 8-day |
| Leaf area index (LAI) | GLASS01A01 | 1 km | 8-day |
| Elevation | GMTED2010 | 15″ | - |





Terra and Aqua MODIS daily 1 km LST products (MOD11A1/MYD11A1, C6) both provide daytime and nighttime LSTs with the spatial resolution of 1 km (Wan et al., 2015).

Three all-sky products from the Global LAnd Surface Satellite (GLASS) products suite (Liang et al., 2013; Liang et al., 2020) were used, including the GLASS 1 km 8-day surface broadband albedo (ALB) product GLASS02A06 (Liu et al., 2013), GLASS 0.05° daily downward shortwave radiation (DSR) product GLASS05B01 (Zhang et al., 2019), and GLASS 1 km 8-
day leaf area index (LAI) product GLASS01A01 (Xiao et al., 2014). For the ALB product, we used black-sky albedo of shortwave (BSA_sw), visible (BSA_vis), and near-infrared (BSA_nir) bands. As radiation products, DSR and ALB determine the shortwave solar radiation received at the surface and the fraction of total radiation reflected and absorbed by the surface, respectively.

The Global Multi-resolution Terrain Elevation Data 2010 (GMTED2010) elevation dataset, downloaded from the United
States Geological Survey (USGS, https://topotools.cr.usgs.gov/gmted_viewer/), was also chosen to estimate $T_a$.

## 3 Methods

The overall framework of this study is shown in the Fig. 2. Firstly, all datasets were pre-processed into identical spatial and temporal resolutions. Second, the values of all datasets were extracted by the nearest neighbour method according to the geographical locations of stations and then matched with the in situ $T_a$ to obtain data pairs. Next, we filled the gaps of MODIS
LSTs and divided all data pairs into three weather conditions according to the gap-filling results. Then, the data pairs under different weather conditions from 2003 to 2016 were randomly divided into training, validation, and test sets (ratio: 3:1:1). Three RF models for different weather conditions were established and trained. The test set was used to validate and evaluate the performance of the $T_a$ estimation models. Finally, we used the models to develop the all-sky $T_a$ dataset and compared it with the existing datasets.







Figure 2. The overall framework of this study.

### 3.1 Data pre-processing

Because the spatial and temporal resolutions of all datasets were not completely consistent, we pre-processed all remotely sensed datasets and reanalysis datasets from 2003 to 2019 into identical 1 km and daily spatial and temporal resolutions, respectively. DSR, elevation and assimilated $T_a$ were resampled to the spatial resolution of 1 km by the nearest neighbour method. As LAI and ALB datasets both have an 8-day temporal resolution, we first combined them into a time series, and then interpolated the time series by linear interpolation method to obtain the daily datasets. For GLDAS assimilation data with a 3-hourly temporal resolution, we averaged all assimilated instantaneous $T_a$ in a day to acquire the assimilated daily mean $T_a$ for all days.

Then, the values of all datasets were extracted by the nearest neighbour method according to the geographical locations of stations and then matched with the in situ $T_a$ to obtain data pairs. Next, we used a temporary gap-filling method to fill the MODIS LST gaps and divided all data pairs into three weather conditions according to the gap-filling results. The detailed





gap-filling method and strategy for weather conditions division is described in the Section 3.2. Then, the data pairs under different weather conditions from 2003 to 2016 were randomly divided into training, validation, and test sets (ratio: 3:1:1).

Among them, training set was used for model training, validation set was used to determine the best model parameters, and test set was used to evaluate the final model performance.

### 3.2 Strategies for LST gap-filling and weather conditions division

MODIS LSTs were produced under strict quality control, with each pixel marked as either a clear-sky or cloudy-sky observation. Pixels under cloudy-sky conditions, had missing LST value, because of which LST-based method could not be

applied to estimate $T_a$. In this study, a simple multi-temporal method was used to fill the MODIS LST gaps. First, we set a time threshold ($\pm$ 2 days), and the missing pixel value was replaced by the clear-sky value of the nearest date within the set time threshold. If no clear-sky pixel was found within the time threshold, the missing pixel was not filled to avoid introducing a high uncertainty caused by a huge temperature change between dates with a large difference. This multi-temporal method was used to fill the gaps of all four MODIS LSTs each day.

Considering the differences in the relationship between $T_a$ and other features under different weather conditions, we divided data pairs into clear-sky conditions and cloudy-sky conditions according to the LSTs gap-filling results. When all four LSTs in a day were all under clear-sky conditions, the data pair was identified as being under clear-sky condition; otherwise, it was identified as being under cloudy-sky conditions. To control the uncertainty introduced by LST gap-filling, cloudy-sky conditions were divided into two cases: Case I and Case II. In particular, a data pair was identified as being under cloudy-sky

conditions case I when there were LST gaps in the data pair and the gaps could be filled through the method mentioned above. If the LST gaps could not all be filled, the data pair was identified as being under cloudy-sky conditions case II. Therefore, we finally divided all data pairs into three types of weather conditions: (1) clear-sky conditions, (2) cloudy-sky conditions case I, and (3) cloudy-sky conditions case II. The detailed criteria for dividing weather conditions are shown in Fig. 3.





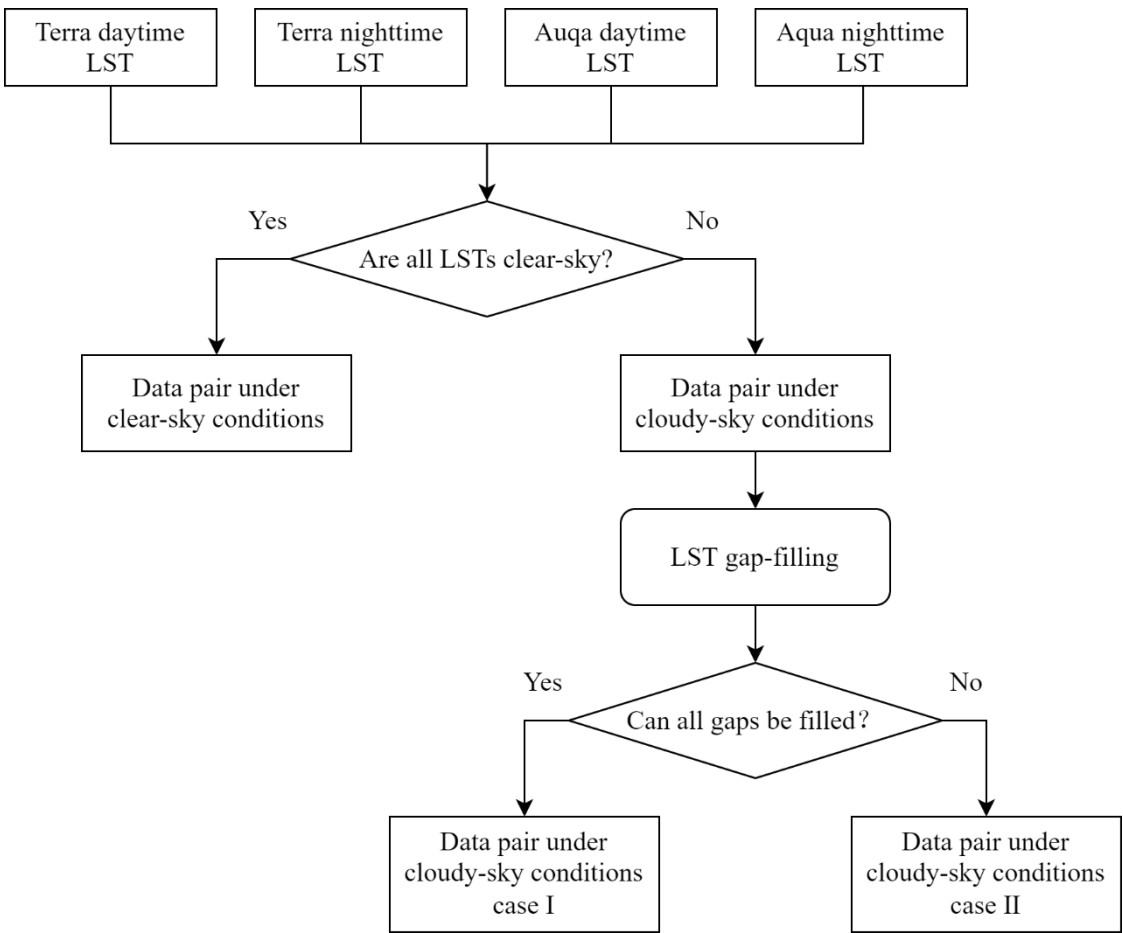

**Figure 3. The criteria for weather conditions division of a data pair.**

Next, we established three machine learning models (clear-sky model, cloudy-sky model I, and cloudy-sky model II) and trained separately for different weather conditions. Daily LSTs were used in models for clear-sky conditions (clear-sky model) and cloudy-sky conditions case I (cloudy-sky model I), but not for cloudy-sky conditions case II (cloudy-sky model II). GLDAS assimilated $T_a$, GLASS DSR, GLASS ALB, GLASS LAI, elevation, and temporal and locational information were 190 also used in all three models as input features. For the clear-sky model, the utilized features included four clear-sky LSTs in a day. The qualification for a pixel of a given day to be judged as clear-sky may be harsh, but this ensured the use of completely clear-sky LSTs. The features of cloudy-sky model I included gap-filled LST(s), which increased the availability of LST, but the simple gap-filling strategy also introduced errors to the models. To avoid instilling a high uncertainty caused by a large temperature change between dates with a large difference, cloudy-sky model II did not use LST to estimate $T_a$.




### 3.3 Random forest

The RF method is an ensemble learning method based on Classification And Regression Tree (CART) proposed by Breiman et al. (1984). Since it was proposed, it has attracted the attention of quite a few fields and been applied to various applications in remote sensing in recent years (Gislason et al., 2006; Ham et al., 2005; Li and Zha, 2019; Xu et al., 2014).

A decision tree is a tree-like prediction model composed of nodes and directed edges. In each internal node of the decision tree, the sample set is segmented by selecting the optimal splitting feature until the segmentation termination condition is reached. Each path from the root node to the leaf nodes of a decision tree forms a classification. There are many algorithms for decision tree, such as ID3 (Quinlan, 1986), C4.5 (Quinlan, 1992), and CART. These algorithms all adopt the top-down greedy algorithm, and each internal node chooses the feature with the best classification effect to split, to achieve the goal of dividing samples into subsets that are as homogenous as possible, with the fastest speed. In the generation algorithms of ID3 and C4.5 decision tree, information gain or information gain rate is used as the criterion to judge the optimal segmentation. Another type of optimal segmentation criterion is Gini impurity, which is utilized in the CART decision tree. In the RF model, multiple CART decision trees are included. The bagging method (Breiman, 1996) is used to generate independent identically distributed training sample sets for each tree and train on them.

Although the application of RF at present is mainly focused on classification, it can be also used in regression analysis effectively, which can usually achieve higher accuracy than traditional regression analysis methods. The training and prediction process of the RF regression model is shown in Fig. 4. First, the bootstrapping method is used to acquire k datasets $\{D_k, k = 1, 2...\}$ and then k decision trees $\{h(x, \Theta_k), k = 1, 2...\}$ are established, respectively, where x is the input vector, and $\Theta_k$ (k = 1, 2...) is the random vector determining the sampling of bootstrap datasets and candidate splitting features of each tree. The construction of a decision tree is realized by iteratively dividing the datasets into two subsets. Different from the RF classification model, the mean square error (MSE) is used as the optimal segmentation criterion in the RF regression model to split the nodes. Each decision tree in the RF regression model takes values rather than types as output targets, and the average of the predicted values of all the trees $\{h(x, \Theta_k), k = 1, 2...\}$ is used as the final prediction.

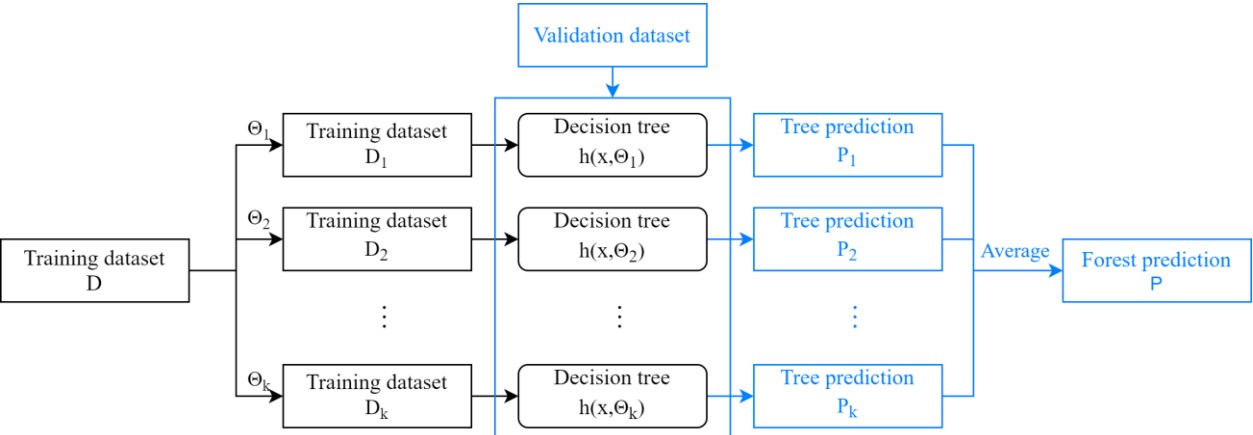

**Figure 4. The training and prediction process of RF regression model.**




### 3.4 Model training and testing


During the model training process, training set was used for model training, validation set was used to determine the models with the optimal hyper-parameters.

Compared with artificial neural network, RF regression model does not need to carry out complicated parameter tuning work and changing some insignificant parameters of the RF model may not cause substantial fluctuations in model performance.

The two most critical hyper-parameters, ntree and mtry, need to be determined during training. Among them, ntree refers to the number of decision trees in the RF model. Increasing ntree is conducive to improving the model performance and stability, but also affects the computational efficiency of the program. Mtry refers to the maximum number of features used in a single decision tree. When mtry is less than the total number of features, the segmentation of a node is determined based on partial features that are randomly selected rather than all features. Increasing mtry allows nodes to consider more features when splitting, but also reduces the diversity of individual trees, thus increasing the risk of overfitting. Therefore, both parameters

need to be properly balanced and selected, and we used the validation set to evaluate the model performance with different combinations of parameters to obtain the optimal hyper-parameters.

Assuming the total number of features of a sample is m, the values of mtry include $\log_2 m$, sqrt(m) and m, and ntree is set to 5-200. To analyse the RF model performance sensitivity to hyper-parameters, the RMSE values of the three models for

different weather conditions were calculated when setting different parameters, and the result is shown in Fig. 5. It can be seen from the results that with the change of model parameters, the three models showed similar variation patterns. With the increase of ntree, the RMSE value decreased gradually until it became almost constant (when ntree ≥ 100). Continue increasing of ntree made very little contribution to improving the model performance but affected the computing efficiency. For mtry, we can see that using partial features (mtry = $\log_2 m$ or sqrt(m)) performed significantly better than using all features (mtry = m). Overall,

setting mtry to $\log_2 m$ and sqrt(m) presented similar performance, and the setting of sqrt(m) performed slightly better than $\log_2 m$ when ntree was larger than 175. Therefore, we set ntree to 200 and mtry to sqrt(m) in all models.

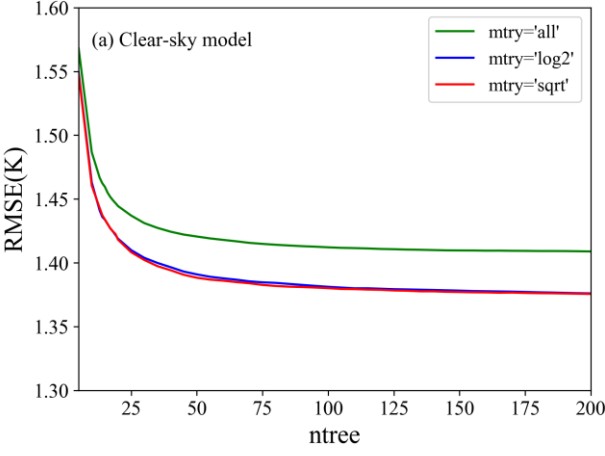
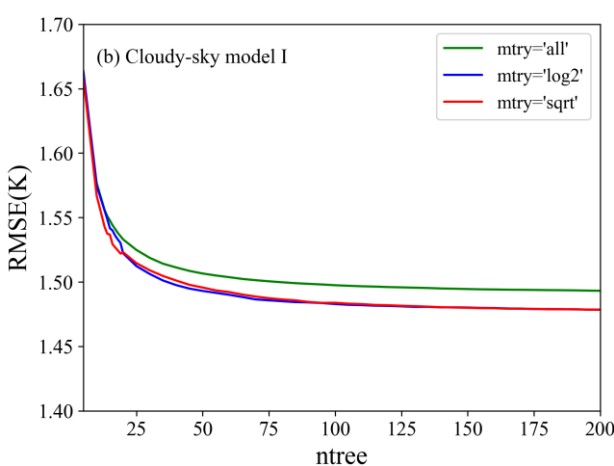

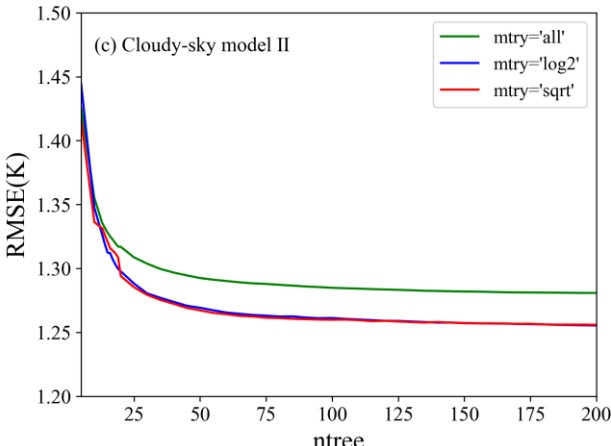

**Figure 5. RF model performance sensitivity to hyper-parameters.**

To quantitatively evaluate the effect of each feature on the models, we calculated the feature importance (FI) of every feature by permutation method for each model. The permutation method breaks the statistical relationship between feature i and the target variable and then measure the degree of deterioration in the model performance to evaluate the importance of feature i to the model (Mcgovern et al., 2019). Specifically, first the model is trained with the training set, and then RMSE of validation set (RMSE$_{true}$) is calculated using Eq. (1). For the calculation of the FI of feature i, RMSE$_i$ is calculated again after all the

features i of validation set are shuffled. The difference between RMSE$_{true}$ and RMSE$_i$ is calculated and then divided by RMSE$_{true}$, and the result is used as FI, as shown in the Eq. (2). A large FI value means that the model performance decreases significantly after shuffling this feature, which indicates that this feature has a great impact on the accuracy of prediction results. On the contrary, if the model performance does not deteriorate significantly, it is obvious that this feature has less influence in the prediction process, or that other linearly dependent features are included in the model to make this feature

redundant.

$$RMSE = \sqrt{\frac{\sum_{i=1}^{n}(y_{pre}-y_{obs})^2}{n}}, \qquad (1)$$

$$FI_i = \frac{RMSE_i - RMSE_{true}}{RMSE_{true}}, \qquad (2)$$

where y$_{pre}$ refers to model prediction result, and y$_{obs}$ refers to the corresponding station observation. RMSE$_{true}$ is the RMSE of the validation set, and RMSE$_i$ refers to the RMSE of the validation set after feature i is shuffled.

To evaluate the performance of the RF models, the prediction results for the test sets were compared with the corresponding station observations. RMSE, MAE and R$^2$ were selected as criteria for model evaluation. The results were grouped by elevation range, land cover type, and month to evaluate the model performance under different situations.



# 4 Results analysis

## 4.1 Overall accuracy and model comparison

Approximately 1/5 of the total data pairs from 2003 to 2016 were randomly selected to evaluate the performance of the final $T_a$ estimation models. Validation statistics of models for different weather conditions and the overall accuracy of all estimated daily mean $T_a$ are shown in Table 2. The three models presented similar validation statistics, with $R^2$, MAE, RMSE, and bias ranging from 0.984 to 0.986, 1.033 K to 1.100 K, 1.342 K to 1.440 K, and 0.012 K to 0.051 K, respectively. The overall $R^2$, MAE, RMSE, and bias of the estimated all-sky $T_a$ were 0.985, 1.068K, 1.409K, and 0.03K, respectively. Compared with the

in situ $T_a$, the estimated $T_a$ of all models showed a high correlation with little difference, confirming the great potential of RF method to estimate all-sky daily mean $T_a$ over a wide spatial and temporal range.

**Table 2. Model validation statistics.**

| Model | $R^2$ | MAE (K) | RMSE (K) | Bias (K) |
|---|---|---|---|---|
| Clear-sky model | 0.986 | 1.033 | 1.342 | 0.021 |
| Cloudy-sky model I | 0.984 | 1.100 | 1.440 | 0.012 |
| Cloudy-sky model II | 0.984 | 1.046 | 1.396 | 0.051 |
| All | 0.985 | 1.068 | 1.409 | 0.030 |

     In addition, to further investigate the distribution of the prediction results and the differences between the three models,

density scatter plots of the estimated $T_a$ against the in situ $T_a$ for the three models are shown in Fig. 6. In the three density scatter plots, most points were very concentrated near the 1:1 line, which also confirmed that these three models have achieved satisfactory accuracy in estimating daily mean $T_a$ under different weather conditions. Among all the models, the clear-sky model had the highest stability and overall accuracy statistically, with the highest $R^2$ and the lowest MAE and RMSE. It could predict $T_a$ under clear-sky conditions from less than 250 K to more than 300 K accurately and steadily. Compared with the

clear-sky model, cloudy-sky model I had a relatively large error, which demonstrated that the LST gap-filling strategy adopted in this study introduced errors into the model to some extent, thereby increasing the uncertainty in estimating $T_a$ under cloudy-sky conditions case I. The accuracy of the cloudy-sky model II was statistically similar to that of the clear-sky model, and it could predict a moderate temperature range close to 275 K with satisfactory performance. However, it can be seen from the density scatter plot for cloudy-sky model II that some discrete points deviated from the 1:1 line in the low-temperature range,

which indicated that there may be much uncertainty in predicting the low-temperature range, especially at temperatures less than 260 K.

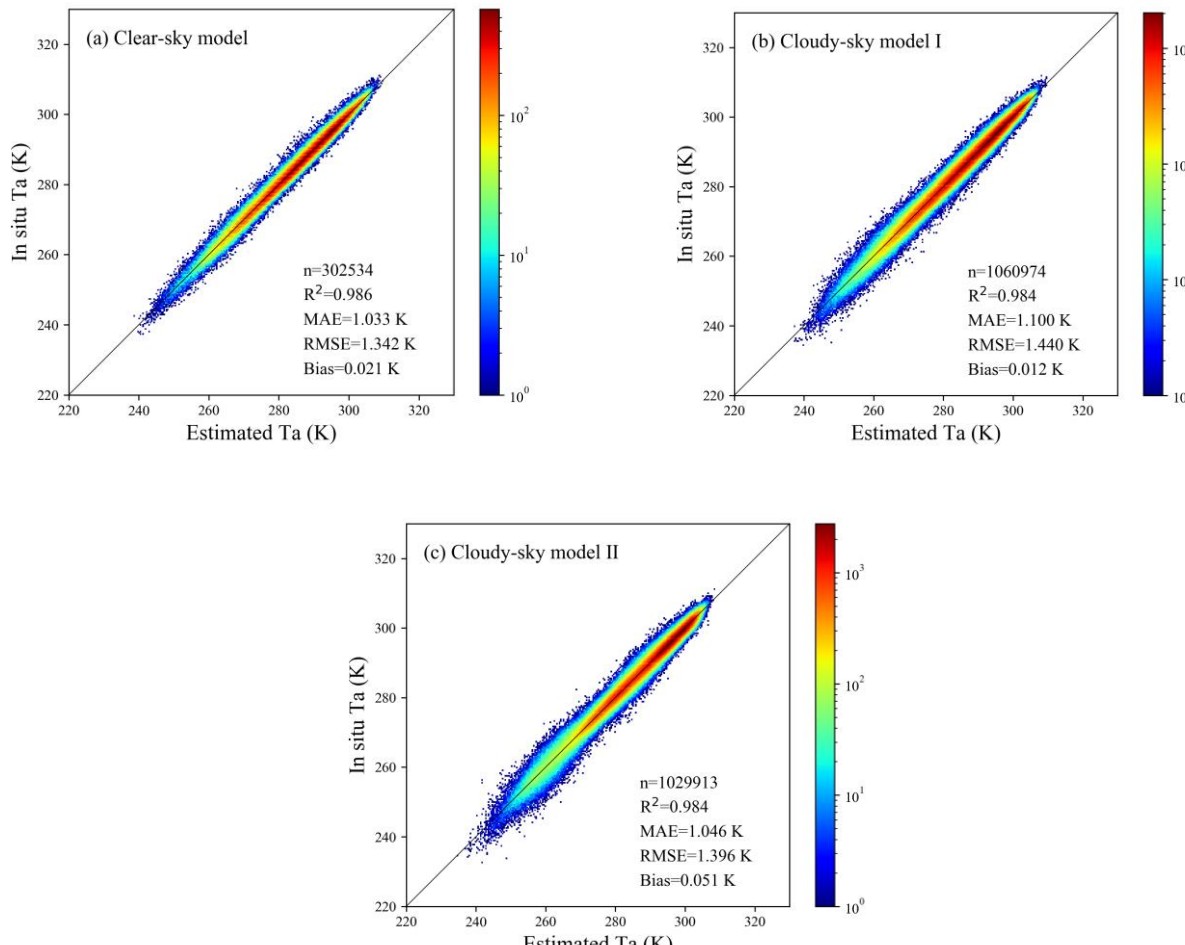

**Figure 6. Density scatter plots of the estimated $T_a$ and in situ $T_a$ for three models.**

Many studies have proved that land cover type and elevation have a significant impact on the heterogeneity of $T_a$ (Benali et al., 2012; Good et al., 2017; Lin et al., 2012; Marzban et al., 2017). Therefore, to comprehensively analyse the performance of the $T_a$ estimation models, we grouped the results by land cover type and elevation range, and then compared the model performance for different groups. The model performance for different land cover types are listed in Table 3. All models showed relatively good performance (RMSE < 1.5 K) for cropland, shrubland, water, and impervious surface while RMSE

values were higher for grassland and bare land, which was consistent with the findings of Shen et al. (2020). The model performance for different elevation ranges is also listed in Table 4. With the increase of elevation, RMSE values of all models had a certain upward trend. However, as shown in the Fig. 7, the elevation of the stations used in this study is mainly distributed in the range 0–2000 m, so the quantity of training samples in this elevation range have an absolute superiority, while the samples of higher elevation (elevation > 2000 m) occupy only a small part. The problem of class imbalance may contribute to





the relatively large errors when predicting $T_a$ at high elevation. In addition, factors such as complex and varied topography, vertical variation in $T_a$, and scale differences between remotely sensed image pixels and station observation data points will lead to high difficulty and uncertainty in $T_a$ estimation at higher elevations (Rao et al., 2019).

**Table 3. Model performance for different land cover types.**

| Land cover type | Clear-sky model | | Cloudy-sky model I | | Cloudy-sky model II | |
|---|---|---|---|---|---|---|
| | % | RMSE (K) | % | RMSE (K) | % | RMSE (K) |
| Cropland | 20.1 | 1.295 | 22.8 | 1.379 | 24.4 | 1.327 |
| Forest | 10.4 | 1.375 | 11.1 | 1.502 | 15.3 | 1.421 |
| Grassland | 26.0 | 1.420 | 22.4 | 1.550 | 17.3 | 1.540 |
| Shrubland | 1.2 | 1.392 | 1.2 | 1.473 | 1.3 | 1.338 |
| Wetland | 0.1 | 1.286 | 0.1 | 1.445 | 0.1 | 2.063 |
| Water | 3.3 | 1.366 | 3.2 | 1.451 | 3.8 | 1.383 |
| Impervious surface | 29.2 | 1.241 | 32.8 | 1.341 | 35.5 | 1.327 |
| Bare land | 9.6 | 1.462 | 6.4 | 1.613 | 2.3 | 1.793 |

**Table 4. Model performance for different elevation ranges.**

| Elevation (m) | Clear-sky model | | Cloudy-sky model I | | Cloudy-sky model II | |
|---|---|---|---|---|---|---|
| | % | RMSE (K) | % | RMSE (K) | % | RMSE (K) |
| < 1000 | 61.8 | 1.281 | 71.1 | 1.381 | 82.4 | 1.363 |
| 1000–2000 | 24.6 | 1.372 | 20.0 | 1.538 | 14.2 | 1.511 |
| 2000–3000 | 6.1 | 1.472 | 4.2 | 1.68 | 1.7 | 1.637 |
| 3000–4000 | 4.7 | 1.547 | 3.0 | 1.619 | 1.1 | 1.614 |
| > 4000 | 2.8 | 1.678 | 1.7 | 1.673 | 0.6 | 1.768 |



Earth System
Science
Data

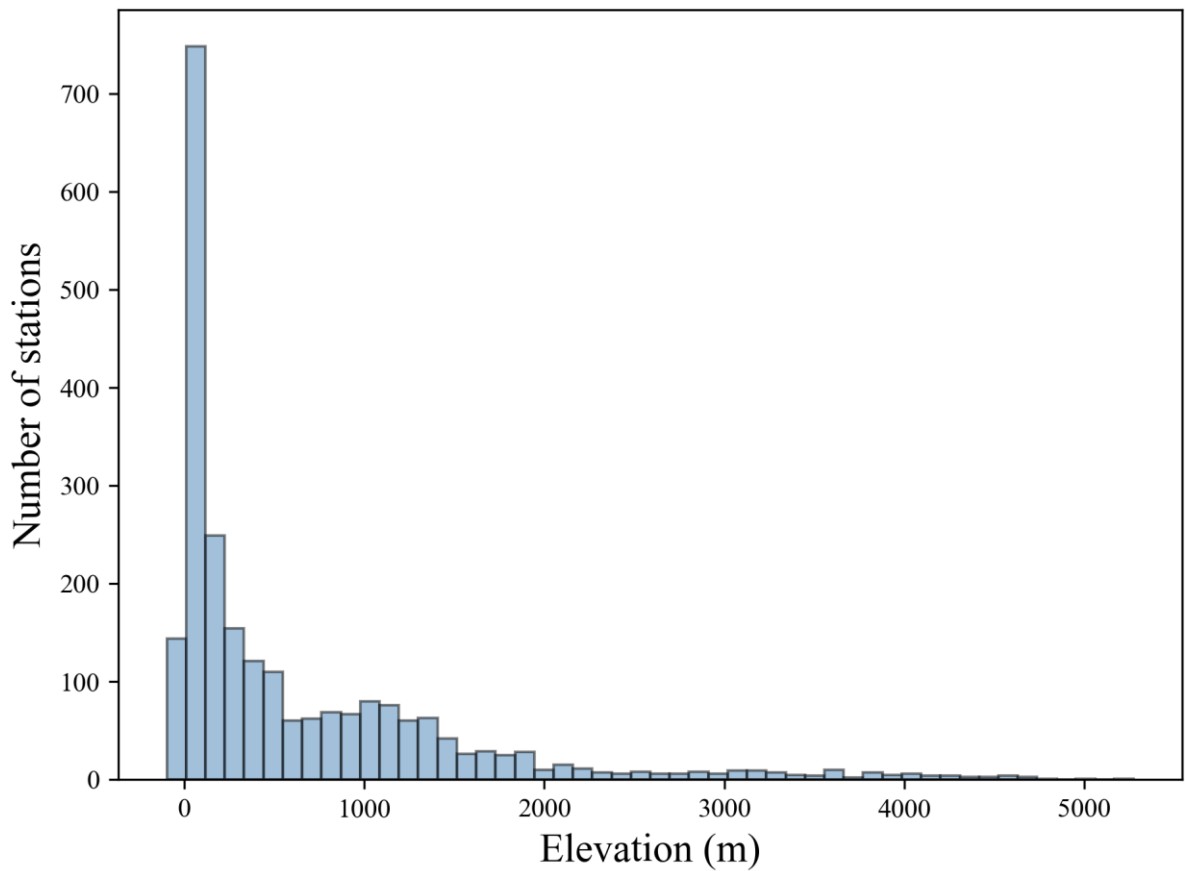

**Figure 7. Elevation histogram of stations used in this study.**

We further evaluated the error distribution of the three models at the stations. Due to the absence of in situ $T_a$ of some ground stations on some days, only the stations that recorded more than 20 days for all three weather conditions were taken to include. And the results of 2320 valid stations were finally obtained, shown in Table 5. In general, the models showed good performance at most stations, with a mean RMSE value of 1.383 K. Moreover, there were 97 % stations with RMSE values less than 2 K and only 1 of the 2320 statistical stations with an RMSE value greater than 3 K. The clear-sky model also had the best performance at the station scale, with the lowest mean RMSE of 1.231 K, and 508 stations had RMSE values less than 1 K, 2286 stations had RMSE values less than 2 K, and only 2 stations had RMSE values greater than 3 K. For the cloudy-sky model I, the mean RMSE reached 1.432 K. RMSE values at 2256 stations were less than 2 K, and only one station had an RMSE greater than 3 K. For the cloudy-sky model II, the mean RMSE was 1.440 K, close to cloudy-sky model I, and 121 stations had RMSE values less than 1 K. However, notably, 13 stations had RMSE values greater than 3 K for cloudy-sky model II, and most of these stations had RMSE values less than 3 K for the other two models.





**Table 5. Error distributions of three models at the stations.**

| RMSE / Model | Mean (K) | < 1 K | < 2 K | < 3 K | ≥ 3 K |
|---|---|---|---|---|---|
| Clear-sky model | 1.231 | 508 | 2286 | 2318 | 2 |
| Cloudy-sky model I | 1.432 | 70 | 2256 | 2319 | 1 |
| Cloudy-sky model II | 1.440 | 121 | 2099 | 2307 | 13 |
| All | 1.383 | 80 | 2249 | 2319 | 1 |

For model comparison, as expected, the clear-sky model that used absolutely clear-sky LSTs performed better than cloudy-sky model I and cloudy-sky model II in almost every aspect and presented the highest stability. Cloudy-sky model I, which contained gap-filled LSTs, did not perform as well as the clear-sky model because although the time threshold (+2 days) of

the LST gap-filling method was relatively small, the LST value of a missing pixel of a date may be replaced by a clear-sky value with a difference of up to 2 days. However, the LST can vary considerably in just a few days, so the LST gap-filling process can introduce large errors into the model, thus affecting the accuracy of $T_a$ estimation. Surprisingly, the cloudy-sky model II that did not use LST features achieved a comparative accuracy with the clear-sky model (RMSE = 1.396 K vs. 1.342 K) statistically. However, when we further analysed the model performance in specific situations, we detected the differences

in the performance of the three models. There may be considerable uncertainty for cloudy-sky model II in predicting the low temperature range, especially at less than 260 K. Notably, the cloudy-sky model II performed poorly on wetlands with an RMSE of 2.063 K, while both clear-sky model and cloudy-sky model I performed well on this type of land cover. This may be because wetlands are a mixture of water and land, with diverse complex ecological environments. Using LST can significantly improve the $T_a$ estimation accuracy of this land cover type.

In summary, because of the strong correlation between $T_a$ and LST, adding daily LSTs as features to models can improve the model stability and robustness. In the absence of LST, assimilated $T_a$ can be used as a substitute for LST to provide an initial value or first guess for the model to estimate $T_a$ with acceptable accuracy when combined with other features. However, the resolution of the reanalysis product is relatively coarse, and some local details were ignored when sampling from a larger scale (0.25°) to a smaller scale (1 km), thus causing certain uncertainties for cloudy-sky model II to predict low-temperature

range or some regions, especially some specific land cover types or regions with complex terrain. Overall, none of the three models showed significant differences in the model performance, and the model performance discrepancies for different land cover types and elevation ranges were acceptable. The proposed models can perform well in different situations and are suitable for $T_a$ estimation under different weather conditions.

## 4.2 Feature importance analysis

To quantitatively evaluate the contribution of each feature included in the RF models, the FI of every feature for the three models was calculated by permutation method described in Section 3.4, and then ranked. To reduce the impact of contingency on the experimental results, we repeated the experiment 30 times and took the average value of all experimental results as the final FI of each feature for each model. The FI results are shown in Fig. 8, with the importance decreasing from top to bottom. The grey line indicates the FI range of each feature for multiple repeated experiments. All features are divided into four types

and represented by different colours, among which the blue rectangles represent MODIS LSTs, the orange rectangles represent GLDAS assimilated $T_a$, the red rectangles represent radiation products including DSR and ALB, and the green rectangles represent other features.

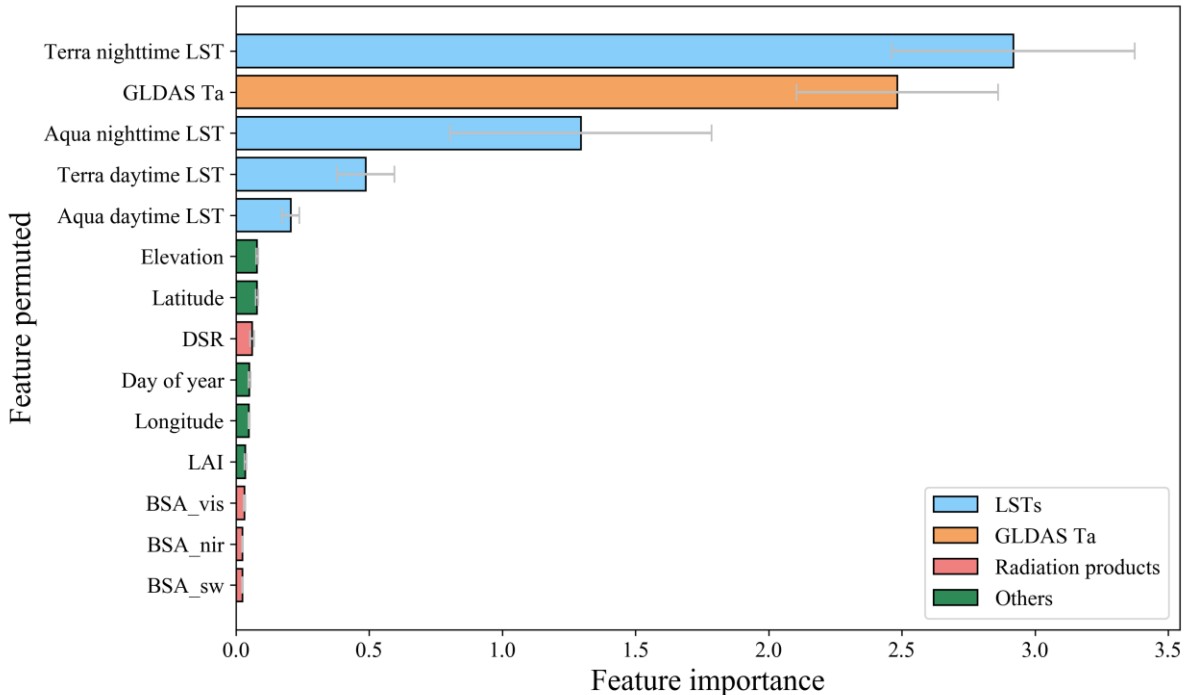

(a) FI of each feature for the clear-sky model.





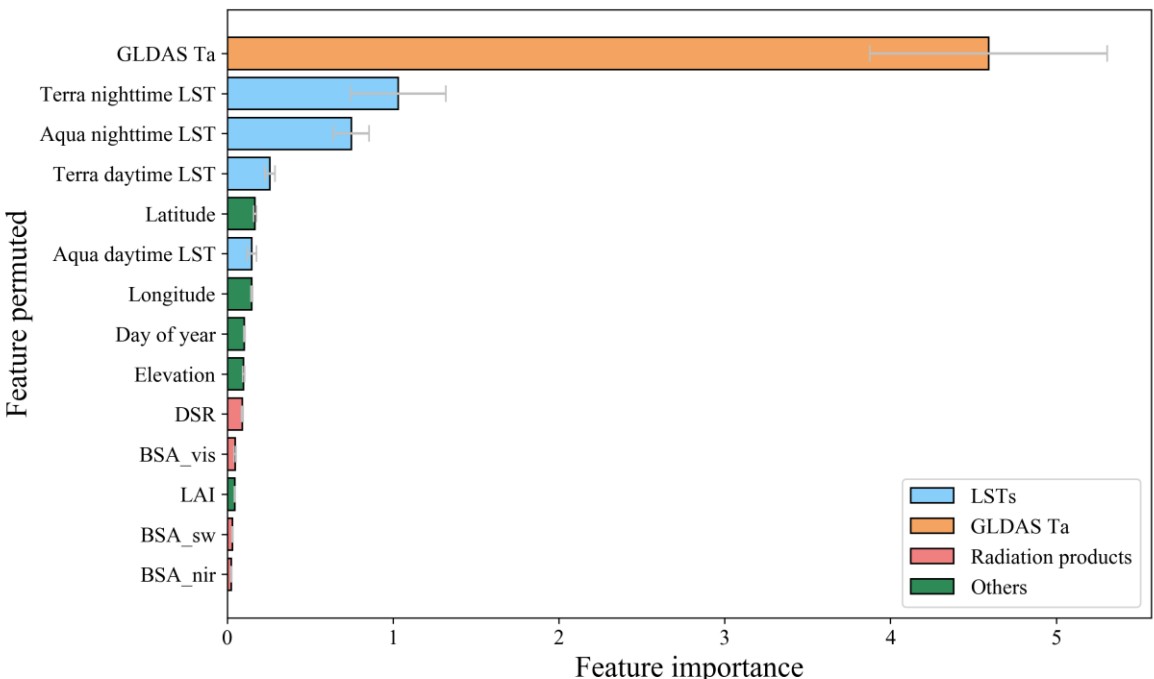


**(b) FI of each feature for the cloudy-sky model I.**

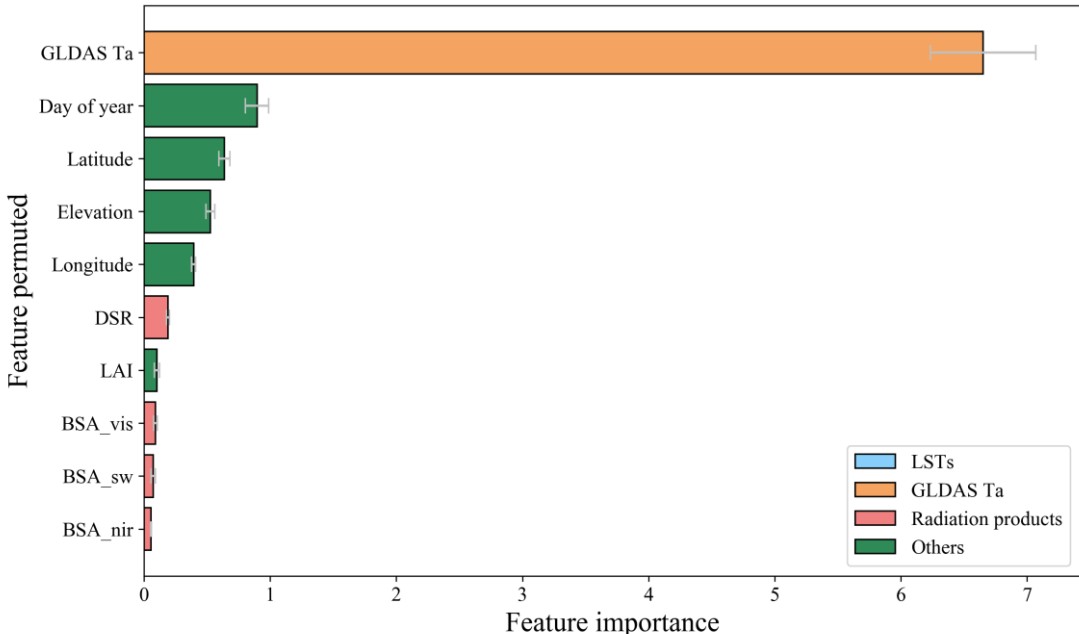

**(c) FI of each feature for the cloudy-sky model II.**

**Figure 8. FI of each feature for three RF models.**



For the clear-sky model, Terra nighttime LST was of the highest importance (FI = 2.92), followed by assimilated $T_a$ (FI = 2.48), indicating that the prediction accuracy of the clear-sky model was significantly reduced after permuting these two features. They were followed by Aqua nighttime LST (FI = 1.3) and two daytime LSTs (FI = 0.49 and 0.21, respectively). For cloudy-sky model I, assimilated $T_a$ ranked first (FI = 4.59), followed by Terra nighttime LST (FI = 1.03). For cloudy-sky model II that did not include LST as features, assimilated Ta played a more importance role (FI = 6.65) than it did for cloudy-

sky model I. The FI of radiation products and other features were all less than 1 for all the models, showing that they only slightly improved the model performance.

The energy exchange between the land surface and the near-surface atmosphere takes the form of longwave radiation, evapotranspiration and turbulent exchange, or other phenomena. LST and land surface emissivity (LSE) determine the longwave radiation in land surface radiation and energy budgets (Liang and Wang, 2019). Thus, there is a strong and

complicated physical correlation between LST and $T_a$. It can be seen from Fig. 8 that all four daily LSTs, especially nighttime LSTs, had relatively high FI for both clear-sky model and cloudy-sky model I. Among all the daily LSTs, nighttime LSTs outweighed daytime LSTs, and Terra nighttime LST was of higher importance than Aqua nighttime LST, which was consistent with the findings of many studies (Benali et al., 2012; Li and Zha, 2019; Zhang et al., 2011). This phenomenon is largely due to the fact that the pass time of Terra was at an approximate local solar time of 10:30 p.m. during the night, when the measured

LST was closer to daily mean $T_a$. In Lin's study, the MAE between LST and $T_a$ during the day and during the night were calculated separately, finding that there was better agreement between LST and $T_a$ during the night (Lin et al., 2012). In addition, because of the lack of solar radiation and its influence on the thermal infrared signal, remotely sensed nighttime LST products usually have higher stability (Benali et al., 2012; Vancutsem et al., 2010).

Assimilated $T_a$ also mattered considerably for $T_a$ estimation models. Its FI was second only to Terra nighttime LST for the

clear-sky model and highest for the cloudy-sky model I and cloudy-sky model II. For cloudy-sky model I, originally missed LSTs were replaced with clear-sky values of a near date, and the error introduced by this simple LST gap-filling strategy resulted in a decrease in the overall LST accuracy, thereby leading the FI of assimilated $T_a$ to exceed that of LSTs. Compared with the cloudy-sky model I, assimilated $T_a$ was of higher importance with a FI of 6.65 for cloudy-sky model II, indicating that it became the absolute dominant factor in $T_a$ estimation when LST was not included in the $T_a$ estimation model. Cloudy-

sky model II also achieved satisfactory accuracy in the validation results. This demonstrates that although the spatial resolution of the assimilated $T_a$ is relatively coarse, it can be the supplement and substitute of MODIS LSTs and provide the initial value or first guess for models to predict $T_a$ with a higher resolution.

Radiation products and other features helped to improve the accuracy of $T_a$ estimation models to a small extent. Among them, latitude, longitude, elevation and day of year had relatively high importance in all three models. Latitude and longitude

determine the relative position of the sun influencing day length, and thus the distribution of total solar radiation the surface receives throughout the year, which in turn affects the patterns of $T_a$ (Benali et al., 2012). Elevation affects how the ground is heated and how much radiation energy is absorbed by the atmosphere, resulting in vertical variations in $T_a$. In addition, the relationship between $T_a$ and LST has great heterogeneity in different regions and at different times and is greatly affected by



surface characteristics and atmospheric conditions. The day of year helps to explain the seasonal changes in atmospheric
physical conditions, chemical composition, and surface characteristics to distinguish the different relationships between $T_a$ and
LST in different seasons and then improve the accuracy of $T_a$ estimation (Yao et al., 2019; Zhang et al., 2011). For LAI, DSR,
and ALB, it is likely that other collinear features in the models made the information provided by them redundant, so their FI
was relatively low in the $T_a$ estimation models.

## 4.3 Spatial distribution of accuracy

The RMSE value was calculated for each meteorological station that recorded more than 20 days for all three weather
conditions. To obtain a deeper understanding of the spatial distribution of model performance, the RMSE spatial distribution
of stations for the three models was mapped, as shown in Fig. 9. It is evident that the model performance varied at different
geographical locations for all three models. The clear-sky model presented the most stable results in different regions compared
with cloudy-sky model I and cloudy-sky model II, with the RMSE values of all stations ranging from 0.566 K to 3.453 K. The
RMSE range of cloudy-sky model I was 0.823–4.370 K, and that of cloudy-sky model II was 0.809–4.198 K. The spatial
patterns of cloudy-sky model I and cloudy-sky model II were generally similar, but for cloudy-sky model II there were more
stations with good performance (RMSE < 1 K) and poor performance (RMSE > 3 K), showing relatively poor stability.

Overall, the stations in central, eastern, and southern China presented high levels of accuracy for all three models, with the
RMSE values of most stations in these places less than 1.5 K. Most stations with large RMSE values were located in southwest,
northwest, and northern China, which was consistent with the results of Shen et al. (2020), and the RMSE values of cloudy-
sky model II in these positions were larger than those of clear-sky model and cloudy-sky model I. On the one hand, the spatial
heterogeneity of model performance is largely because of the uneven distribution density of meteorological stations. As can
be seen from the geographical locations of the meteorological stations used in this study in Fig. 1, it is obvious that stations in
central, eastern, and southern China are densely distributed, while stations in northern and western China are relatively rare,
which may contribute to the uneven distribution of model performance. Additionally, the terrain environment in central, eastern,
and southern China is not complex, while high elevation and some climate types will increase the uncertainty of $T_a$ estimation
in northern and western China. The climate types of stations with poor performance were mostly temperate continental and
plateau mountain climates, and the land cover types were mainly bare land and grassland. It can be seen from Table 4 that
cloudy-sky model II showed relatively poor performance for these two land cover types. Therefore, there was a certain
uncertainty when only assimilated $T_a$ and other features except LSTs were included to predict $T_a$ in places with these climate
and land cover types. Overall, although the spatial distribution of the model performance was relatively uneven, the $T_a$
estimation models for different weather conditions all showed satisfactory performance.



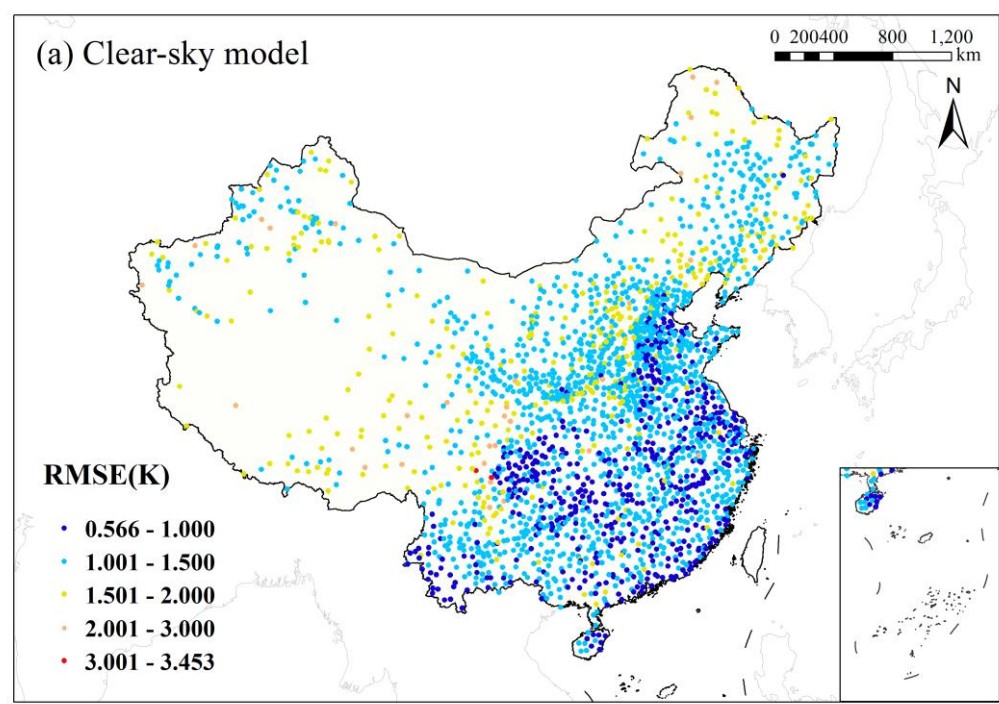


(a) RMSE spatial distribution for clear-sky model.

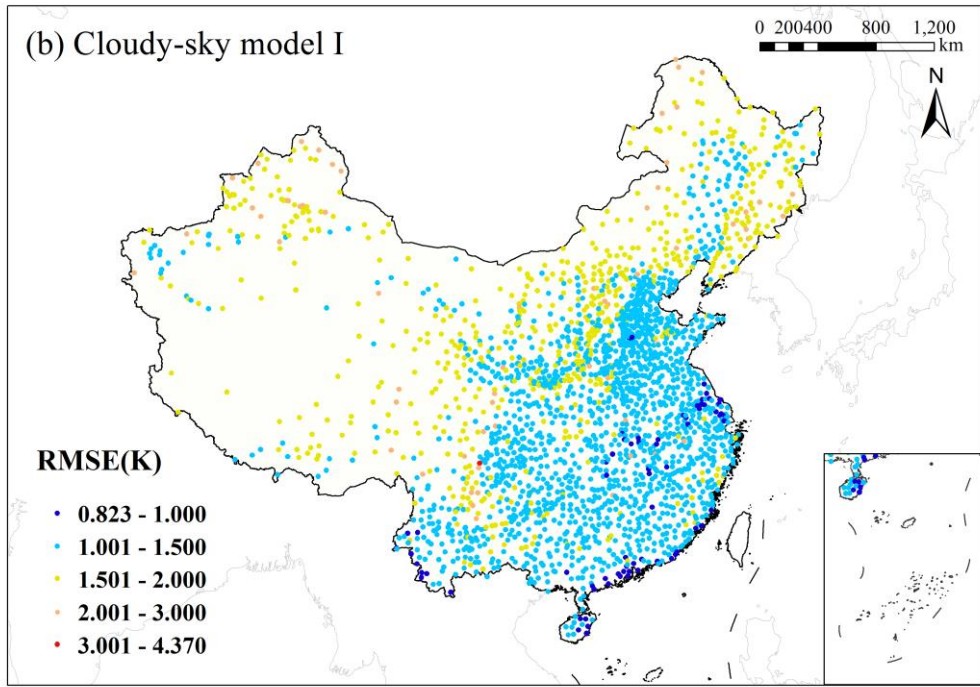

(b) RMSE spatial distribution for cloudy-sky model I.





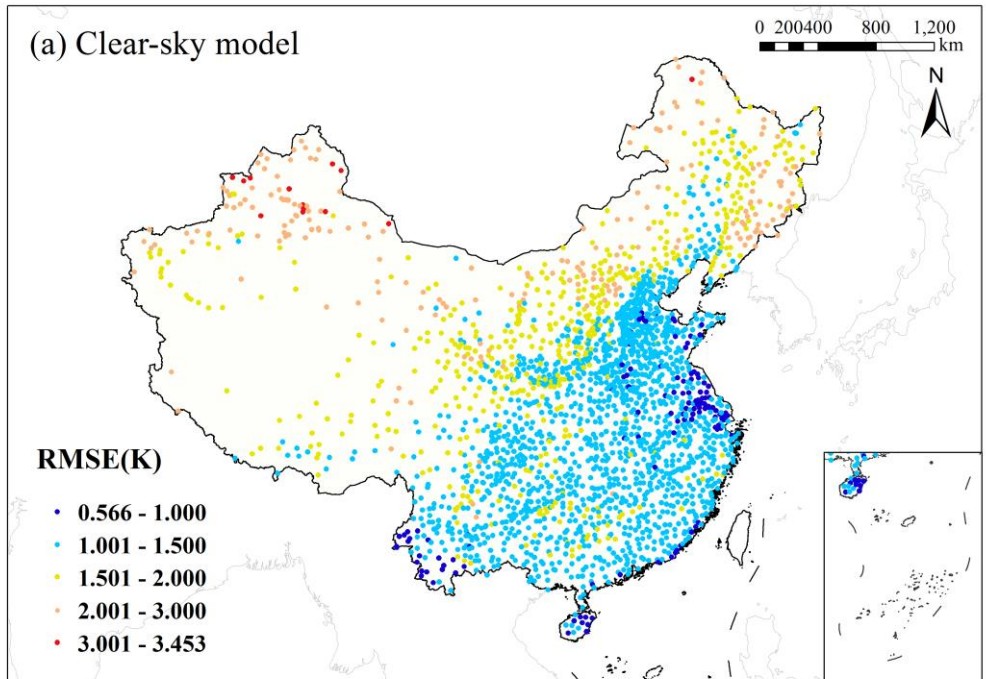

**(c) RMSE spatial distribution for cloudy-sky model II.**

**Figure 9. RMSE spatial distribution of stations for three RF models.**

**4.4 Seasonal distribution of accuracy**

The model performance at the monthly scale was also evaluated, and the RMSE monthly distribution for the three models is shown in Fig. 10. The RMSE range of the clear-sky model was 1.109–1.508 K, cloudy-sky model I was 1.178–1.692 K, and

cloudy-sky model II was 1.056–1.777 K. It is obvious that there was temporal heterogeneity in the model performance, and the estimation accuracy presented similar seasonal variation patterns for all three models. The RMSE values were lower in summer and autumn, and higher in spring and winter, reaching a peak in February and reaching a bottom in July or August. We can conclude that models performed better in warm days, with the RMSE values of all three models below 1.22 K in July and August. This finding was consistent with the validation results at the monthly scale of Yao et al. (2019) and Li and Zha

(2019). This phenomenon may be partly due to the fact that China is vast in territory with a latitudinal difference between the northernmost station and the southernmost stations of about 30°, so the range of $T_a$ is wider in cold days than in hot days.

Monthly differences in model performance also indicated that the relationship between $T_a$ and other factors varied seasonally and may have been more consistent in the same month. It was confirmed in the research of Yao et al. (2019) that modeling data of the same month together could achieve more accurate results. Therefore, although day of year was used in the modeling

in this study, this temporary difference was not completely eliminated. Modeling the datasets of all seasons together in this



study may increase the temporary heterogeneity of accuracy. It is worthwhile to consider grouping the data of the same month to establish monthly models in the future, which may be conducive to further improving the accuracy of $T_a$ estimation.

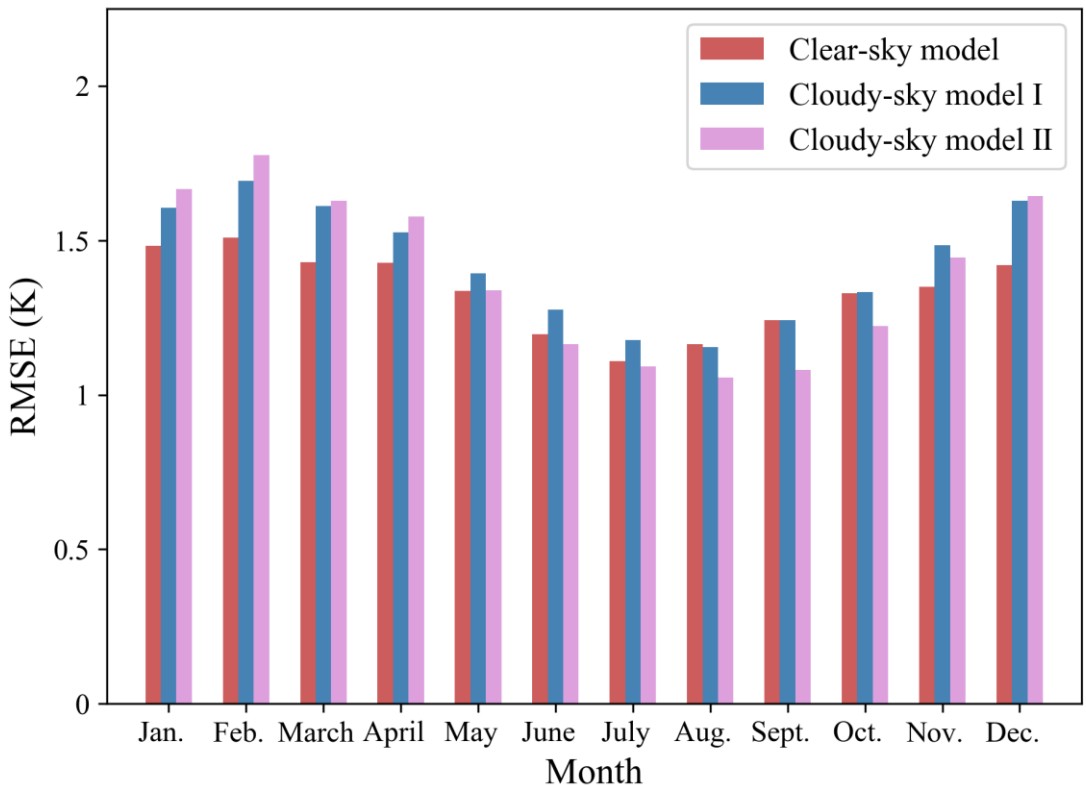

**Figure 10. RMSE monthly distribution for three RF models.**

**5 Comparison with existing datasets**

For a more comprehensive evaluation of the estimated daily mean $T_a$, we compared it with three reanalysis and meteorological forcing datasets including CLDAS, CMFD, and GLDAS in terms of validation statistics and spatiotemporal patterns. The station observations in 2010 were used to validate the accuracy of these four $T_a$ datasets. It should be noted that we estimated daily mean $T_a$ for the period ending at local midnight rather than 24:00 UTC. To ensure the time consistency, we calculated

the average value of all simulations on a local day as the daily mean $T_a$ for the reanalysis and meteorological forcing datasets. The statistical results and the density scatter plots are shown in Table 6 and Fig. 11, respectively. It can be seen that compared with the reanalysis datasets, the RF $T_a$ presented the highest consistency with the station observations, with the best performance in all accuracy assessment criteria ($R^2$, MAE, RMSE, and bias values were 0.992, 0.680 K, 1.010 K, and 0.063 K, respectively). The points in the density scatter plot of the RF $T_a$ were more concentrated near the 1:1 line. CLDAS $T_a$ and

CMFD $T_a$ both showed near zero bias with the station observations, but their RMSE values were both close to 2 K. GLDAS



$T_a$ reported slight underestimation (bias = 0.900 K). In general, this comparison confirmed the applicability of RF method in $T_a$ estimation and the higher accuracy of our estimated $T_a$ compared to the reanalysis products.

**Table 6. Evaluation results of four datasets in 2010.**

| $T_a$ | $R^2$ | MAE (K) | RMSE (K) | Bias (K) |
|---|---|---|---|---|
| RF | 0.992 | 0.680 | 1.010 | 0.063 |
| CLDAS | 0.972 | 1.427 | 1.938 | -0.078 |
| CMFD | 0.962 | 1.642 | 2.242 | 0.092 |
| GLDAS | 0.938 | 2.160 | 2.874 | 0.900 |

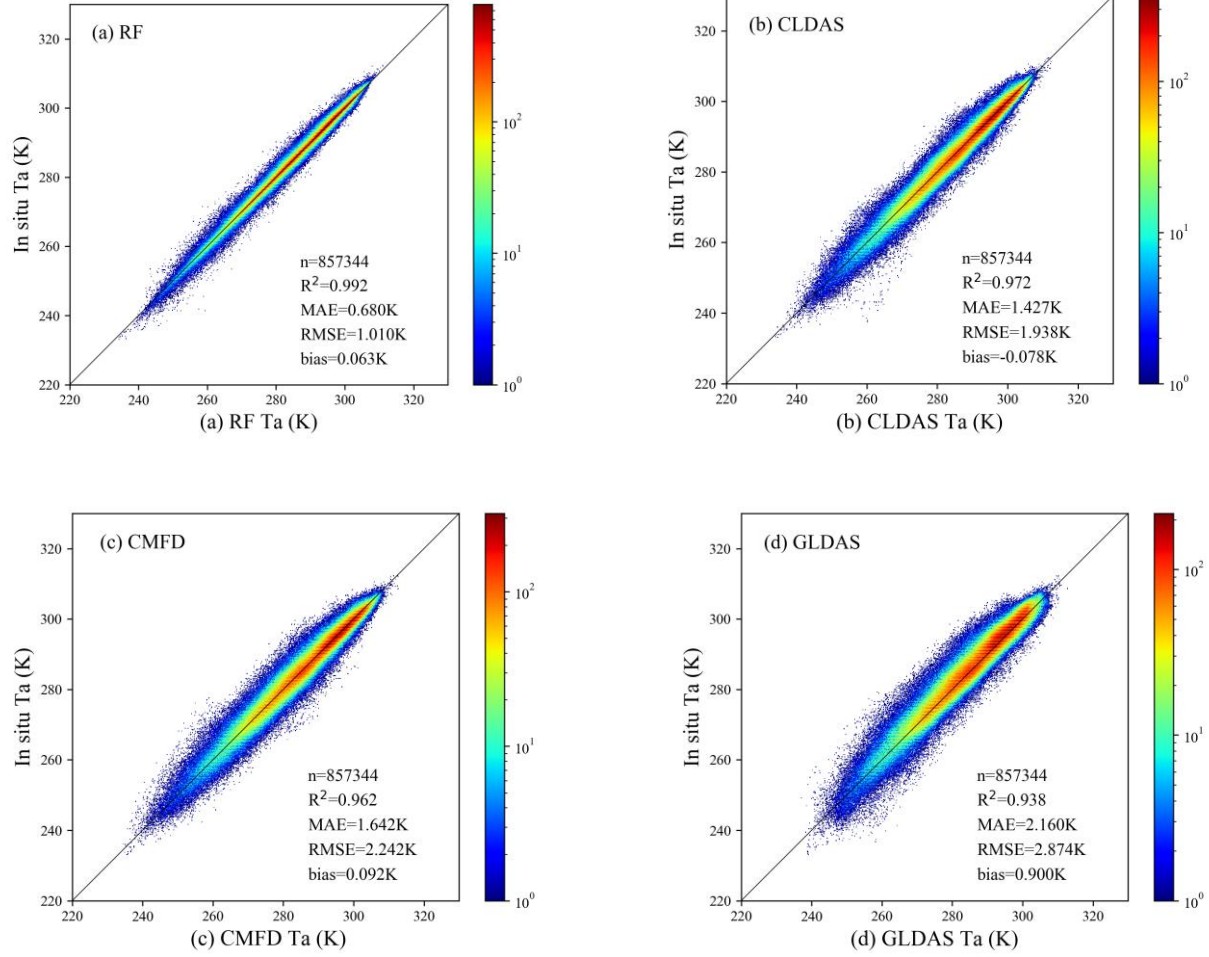


**Figure 11. Density scatter plots of the estimated $T_a$ and reanalysis $T_a$ against the station observed $T_a$ in 2010.**





In addition, the spatiotemporal patterns of these four $T_a$ datasets were compared. We calculated the monthly mean $T_a$ in 2010 for all datasets, and the RF monthly mean $T_a$ mappings over mainland China in February, May, August, and November

2010 are shown in Fig. 12 (a–d). The CLDAS (Fig. 12 (e–h)), CMFD (Fig. 12 (i–l)) and GLDAS (Fig. 12 (m–p)) monthly mean $T_a$ mappings in the same months are also shown in Fig. 12. The spatial resolutions of RF, CLDAS, CMFD and GLDAS monthly mean $T_a$ are approximately $0.01° \times 0.01°$, $0.0625° \times 0.0625°$, $0.1° \times 0.1°$, and $0.25° \times 0.25°$, respectively. We used GLDAS assimilated $T_a$ and GLASS LAI products in $T_a$ estimation, which have no value in most water bodies, so $T_a$ of these areas was also not estimated.

As can be seen from Fig. 12, it's clear that these four datasets basically showed a high degree of consistency in the spatiotemporal patterns over mainland China overall. China has a vast territory and its topography is high in the west and low in the east. The spatial patterns of $T_a$ over mainland China present great seasonal heterogeneity. In winter, the sun shines directly in the southern hemisphere and the northern hemisphere receives less solar energy consequentially. The $T_a$ in northern China and Tibetan Plateau are generally low, and the $T_a$ difference between the north and the south exceeds 50 K. On the

contrary, in summer, as the sun shines directly in the northern hemisphere, $T_a$ in most parts of China are generally high except for the Tibetan Plateau, with little $T_a$ difference between the north and the south. As an expectable consequence of higher spatial resolution, the RF $T_a$ mappings were capable of providing more details about the $T_a$ spatial patterns than the reanalysis and meteorological forcing $T_a$, especially in mountainous areas with complicated terrain. GLDAS $T_a$ presented an obvious pixel effect because of the relatively coarse spatial resolution. In summary, the all-sky daily mean $T_a$ product developed in this

study has achieved satisfactory accuracy and high spatial resolution simultaneously, which can reveal the seasonal variation trend and the spatial patterns of $T_a$ over China well. This product can provide a long time series of daily mean $T_a$ with the spatial resolution of 1 km over mainland China, which fills the current dataset gap in this field. Moreover, this product is also conducive to observing and analysing the climate characteristic of China and plays an important role in the studies of climate change and hydrological cycle.













**Figure 12. Mappings of monthly mean $T_a$ over mainland China. (a–d) are the RF $T_a$, (e–h) are the CLDAS $T_a$, (i–l) are the CMFD $T_a$, (m–p) are the GLDAS $T_a$ in February, May, August, and November 2010, respectively. The white pixels in mainland China indicate no data value, which are always water bodies.**

## 6 Data availability

The daily mean $T_a$ product over mainland China is freely available at http://doi.org/10.5281/zenodo.4399453 (Chen et al., 2021b) from 2003 to 2008 and at the University of Maryland (http://glass.umd.edu/Ta_China/) from 2003 to 2019 currently. In order to make this big dataset easier to understand and use, we made a provincial sub-dataset with a smaller geographic coverage. An all-sky 0.01° daily $T_a$ product over Beijing (2003–2019) was generated from the developed dataset after resampling and clipping, and it is publicly available at http://doi.org/10.5281/zenodo.4405123 (Chen et al., 2021a).

The MODIS product and GLDAS dataset were downloaded via the website https://earthdata.nasa.gov/. The GLASS products were downloaded at www.glass.umd.edu. The CLDAS dataset and CMFD dataset were downloaded at http://tipex.data.cma.cn and the website http://data.tpdc.ac.cn/, respectively.



## 7 Conclusion

$T_a$ is a key variable in climate and global change research. In this study, we developed an all-sky 1 km daily mean $T_a$ product for 2003–2019 over mainland China mainly based on MODIS and GLDAS data using the RF method. An efficient temporary gap-filling method was first used to fill MODIS LST gaps under cloudy-sky conditions. We predicted $T_a$ under three different weather conditions separately: clear-sky conditions (when the daily LSTs are all clear-sky), cloudy-sky conditions case I (when the daily LST gap(s) can be filled), and cloudy-sky conditions case II (when the daily LST gap(s) cannot all be filled). The validation results using station measurements (1/5 of the total data from 2003 to 2016 selected randomly), which were not used for model training, showed that the $R^2$ values were 0.986, 0.984, and 0.984, RMSE values were 1.342 K, 1.440 K, and 1.396 K for clear-sky model, cloudy-sky model I, and cloudy-sky model II, respectively. In general, the models showed excellent performance at most stations, with a mean RMSE of 1.383 K, and there were 97 % stations with RMSE values less than 2 K and only 1 of 2320 stations with an RMSE value greater than 3 K. In addition, we examined the spatial and temporal patterns of the model accuracy and dependence on the land cover types and concluded that model performance under all conditions was acceptable overall, despite some heterogeneity under different conditions. The relative contributions of different features to models were also quantitatively analysed, and it was found that LST and assimilated $T_a$ were of great significance in $T_a$ estimation. Finally, we compared the estimated $T_a$ in 2010 with CLDAS, CMFD, and GLDAS datasets. The estimated $T_a$ in this study showed great consistency in the spatiotemporal patterns with these three datasets and reported significantly higher accuracy against the station observations. The $R^2$, RMSE, and bias values of the estimated $T_a$ in 2010 were 0.992, 1.010 K, and 0.063 K, respectively.

Overall, this study developed a robust scheme to use a machine learning method to estimate all-sky daily mean $T_a$ over a large spatial and temporal range. This approach can be applied globally. The generated all-sky $T_a$ product had achieved a high degree of accuracy compared with the existing datasets, which fills the current dataset gap in this field and plays an important role in many scientific fields such as climate change, hydrological cycle, and energy balance. Future work should focus on developing better LST gap-filling methods, experimenting with more advanced deep learning methods that take into account the spatial and temporal dependence of $T_a$.

## Author contributions

SL and YC contributed to the design of this study and developed the overall methodology. HM, BL, and YC collected and pre-processed the data. YC carried out the experiments. YC, BL, TH, and QW produced the product. YC wrote the first draft. All authors revised the manuscript.



**Competing interests**

All authors declare that they have no conflicts of interest.

**Acknowledgements**

We gratefully acknowledge the data support from "National Earth System Science Data Center, National Science & Technology Infrastructure of China (http://www.geodata.cn)". We thank the GLASS team for providing the data used in this study, which can be downloaded at www.glass.umd.edu. We are grateful to the National Aeronautics and Space Administration team for providing the MODIS product and GLDAS data freely download via the website https://earthdata.nasa.gov/. We also

thank the CLDAS and CMFD teams for providing available CLDAS datasets and CMFD datasets freely download via the website http://tipex.data.cma.cn and the website http://data.tpdc.ac.cn/, respectively. Additionally, authors would like to acknowledge the Chinese Meteorological Administration for providing available in situ measurements.

**Financial support.**

This study was partially supported by the Chinese Grand Research Program on Climate Change and Response under the project

2016YFA0600103.

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
