# Peer review of "An all-sky 1 km daily land surface air temperature product over mainland China for 2003–2019 from MODIS and ancillary data"

_Earth System Science Data, 2021_

## Author Comment (AC2)

**Response to Referee #1 Comments**

We thank Referee #1 for the valuable and constructive comments on our manuscript. A point-by-point response to all comments is listed below.

**Point 1:** I'm wondering whether the data from 2003-2016 or 2003-2019 is used and produced. There seems to be inconsistency in the paper regarding the temporal period of the study.

**Response 1:** Thank you for your comments. We used data from 2003 to 2016 for model training and validation, and generated datasets from 2003 to 2019 using the trained models. Specifically, the data pairs from 2003 to 2016 were randomly divided into training, validation, and test sets (ratio: 3:1:1). Among them, training set was used for model training, validation set was used to determine the best model parameters, and test set was used to evaluate the final model performance. After model training, we used the models to develop the all-sky $T_a$ dataset from 2003 to 2019. We have added the details on page 7, lines 146-156 in the revised manuscript:

145 **3 Methods**

The overall framework of this study is shown in the Fig. 2. Firstly, all datasets from 2003 to 2019 were pre-processed into identical spatial and temporal resolutions. Second, we filled the gaps of MODIS LSTs and then divided all data pairs into three weather conditions according to the gap-filling results.  Next, the values

150 of all datasets were extracted by the nearest neighbour method according to the geographical locations of stations and then matched with the in situ $T_a$ to obtain data pairs.  We used data pairs from 2003 to 2016 for model training and validation, and generated datasets from 2003 to 2019 using the trained models. data pairs under different weather conditions from 2003 to 2016 were randomly divided into training, validation, and test sets (ratio: 3:1:1). Three RF models for different weather

155 conditions were established and trained. The test set was used to validate and evaluate the performance of the $T_a$ estimation models. Finally, we used the models to develop the all-sky $T_a$ dataset from 2003 to 2019 and compared it with the existing datasets.

**Point 2:** For vadiation of the study, how is the performance of the dataset/model if validation is carried out using a time period different from training period? For example, training is done using data from 2003 to 2016 and validation is done using data from 2017-2019? This is to see whether the training coeffients or RF models can be used after Terra/Aqua fail in the future.

**Response 2:** Thank you for your comments. We trained the models with the training set from 2003 to 2016, and further evaluated the models with data pairs from 2017 to 2019, which was not used for model training at all. The overall $R^2$, MAE, RMSE, and bias of the validation set were 0.982, 1.233 K, 1.611 K, and -0.340 K, respectively. The RMSE was slightly higher for the validation results using data from 2017 to 2019 compared to the validation results using the test set from 2003 to 2016 (1.611 K vs.

1.409 K). We found that there were certain differences in the $T_a$ distribution between the two time periods. And the difference in the data distribution between the training set and the validation set may result in a slight decrease in the performance of the machine learning models on the validation set. Considering the data distribution range of $T_a$, we consider a difference of about 0.2 K to be acceptable. In general, the RF models have good generalization ability and can predict $T_a$ of other years that have not been learned at all with satisfactory accuracy. We have added the content on page 19, lines 354-369 in the revised manuscript:

**4.2 Independent validation**

355    We further evaluated the models with data pairs from 2017 to 2019, which was not used for model training at all. The overall $R^2$, MAE, RMSE, and bias of the estimated all-sky $T_a$ were 0.982, 1.233 K, 1.611 K, and -0.340 K, respectively. Figure 8 shows the density scatter plots of the estimated $T_a$ against the in situ $T_a$ from 2017 to 2019 under all weather conditions. The RMSE was slightly higher for the validation results using data from 2017 to 2019 compared to the validation results using the test set from 2003 to 2016 (1.611 K vs. 1.409 K). The histograms of $T_a$ data distribution for 2003–2016 and 2017–2019 are

360    shown in Fig. 9 (a) and (b), respectively. Obviously, there are differences in the data distribution of $T_a$ between the two time periods. The $T_a$ from 2003 to 2016 was slightly higher, with a more pronounced peak at around 295 K in the data distribution histogram. The difference in the data distribution between the training set and the validation set may result in a slight decrease in the performance of the machine learning models on the validation set. Considering the data distribution range of $T_a$, a difference of about 0.2 K in RMSE is acceptable. In general, after learning from a sufficiently large training set, the RF models

365    have good generalization ability and can predict $T_a$ of other years that have not been learned at all with good accuracy.

[Figure]

**Figure 8. Density scatter plots of the estimated $T_a$ and in situ $T_a$ of independent validation results.**

[Figure]

**Figure 9. $T_a$ data distribution for 2003–2016 (a) and 2017–2019 (b).**

**Point 3:** I suggest to redo Figure 1 showing the number of data pairs and land types at these stations. You could use the color or the size of the symbol to provide such information.

**Response 3:** Thank you for your comments. We redid Figure 1 in the manuscript to show the spatial distribution and land cover types of the stations, as shown in Fig. 1 below. Each dot represents a station, and different colors correspond to different land cover types as shown in this figure legend. The land cover data used in the study is Finer Resolution Observation and Monitoring of Global Land Cover (FROM-GLC) version2 (2015_v1), which is a 30 m resolution global land cover maps (Gong et al., 2013). We have changed Figure 1 on page 6, lines 128-130 in the revised manuscript:

[Figure]

Figure 1. Study area and the location of meteorological stations used in this study. Each dot represents a station, and different colors correspond to different land cover types as shown in this figure legend.

[Figure]

[Figure]

Figure 1. Study area and the location of meteorological stations locations used in this study. Each dot represents a station, and different colors correspond to different land cover types as shown in this figure legend.

130

We also calculated the number of data pairs from 2003 to 2016 for each station. Figure 2 below shows the number of data pairs of meteorological stations. Because station measurement data or satellite data or assimilation data were missing at some stations on some days, not all stations have data pairs equal to the total number of days. All 2384 meteorological stations used in this study have data pairs ranging from 1091 to 5113 over a 14-year period from 2003 to 2016. There were 2290 stations with data pairs greater than 5000, and only 6 stations with data pairs less than 3000. Overall, there is little difference in the number of data pairs at the station. Further combined with the analysis of the spatial distribution of model accuracy in Section 4 of the manuscript, it is concluded that the number of data pairs has no significant effect on the accuracy of $T_a$ estimation.

[Figure]

Figure 2. The spatial distribution of the number of data pairs from 2003 to 2016 of meteorological stations.

**Point 4:** Could you show the accuracy of the results as a joint function of surface types and surface temperature?

**Response 4:** Thank you for your comments. The relationship between land surface temperature (LST) and error under 8 surface types is represented by different colors as shown in the legend in Fig. 3. The abscissa is the average of the four daily LSTs for a data pair, and the ordinate is the error, which is the difference between the estimated $T_a$ and the station measured $T_a$.

As can be seen from Fig. 3, for different surface types, the number of data pairs and the range of LST are different. The error range is also different. For each surface type, the errors showed no significant difference at different LST, and all present a

normal distribution centered on 0 K. Therefore, the model performance varies with the surface types to some extent, but the estimation accuracy has no significant joint correlation with surface types and LST.

[Figure]

Figure 3. The relationship between LST and error under different surface types.

**Point 5:** If the FI factors are small for surface radiation measurements, why not remove them from your model?

**Response 5:** Thank you for your comments. The radiation features help to reflect the heat exchange process between the surface and the atmosphere. In our experiment, we found that the FI factors of radiation features were small for the $T_a$ estimation models. Table 1 lists the validation results for models with and without radiation features. It can be seen that, after removing DSR and ALB features, the overall RMSE values of the validation set for the three models increased by 0.02-0.06 K. Therefore, the radiation features have little influence on the overall accuracy of the models.

However, in the analysis of the results of some stations, it is found that the accuracy of the models including radiation features was higher than that of the models excluding radiation features at some stations. For example, Fig. 4 below shows the $T_a$ annual curves of four stations in 2010. In the figure, the orange lines are the station measured $T_a$, while the green and blue lines are the $T_a$ predicted by the models with and without radiation features, respectively. RMSE1 and RMSE2 are RMSE values for models with and without radiation features, respectively. The results showed that on some days, adding radiation features to the models helped improve the $T_a$ estimation accuracy at

some stations. Although there may be other collinear features in the models that make the information provided by them redundant, the radiation features can play a supplementary role in the case of some other features that do not perform well. Therefore, we finally decided to retain the radiation features in the $T_a$ estimation models.

Table 1. Validation results for models with and without radiation features.

| Model | Include radiation features | | Not include radiation features | |
|---|---|---|---|---|
| | $R^2$ | RMSE (K) | $R^2$ | RMSE (K) |
| Clear-sky model | 0.986 | 1.342 | 0.985 | 1.365 |
| Cloudy-sky model I | 0.984 | 1.440 | 0.984 | 1.468 |
| Cloudy-sky model II | 0.984 | 1.396 | 0.983 | 1.451 |
| All | 0.985 | 1.409 | 0.984 | 1.448 |

[Figure]

[Figure]

[Figure]

[Figure]

Figure 4. $T_a$ annual curves of station 51334, station 54273, station 54279, and station 56434 in 2010. The orange lines are the station measured $T_a$, while the green and blue lines are the $T_a$ predicted by the models with and without radiation features, respectively. RMSE1 and RMSE2 are RMSE values for models with and without radiation features, respectively.

We have added the reason for retaining the radiation features on page 23, lines 424-427 in the revised manuscript:

> was relatively low in the $T_a$ estimation models. However, in the analysis of the results of some stations, it is found that adding
> 425 radiation features to the models helped improve the $T_a$ estimation accuracy at some stations on some days. The radiation features can play a supplementary role in the case of some other features that do not perform well. Therefore, we finally decided to retain the radiation features in the $T_a$ estimation models.

**Point 6:** There are places in the paper using "temporary gap filling model", but it should be "temporal" instead of "temporary".

**Response 6:** Thank you for your comments. We have modified the words on page 4, line 112, and page 8, line 169 and page 26, lines 474-475, and page 33, line 543 in the revised manuscript:

> The main objective of this study is to develop an all-sky 1 km daily mean $T_a$ over mainland China for 2003–2019 by
> 110 integrating satellite data products, model simulations, and ground measurements. For the first time, assimilated $T_a$ was applied to supplement and substitute MODIS LSTs and provide the initial values of model prediction. In order to solve the issue of missing LST, a simple temporal filling method was used to fill the gaps of MODIS LSTs first. Considering the

Then, the values of all datasets were extracted by the nearest neighbour method according to the geographical locations of stations and then matched with the in situ $T_a$ to obtain data pairs. Next, we used a temporaltemporary gap-filling method to fill the MODIS LST gaps and divided all data pairs into three weather conditions according to the gap-filling results. The detailed

170

Monthly differences in model performance also indicated that the relationship between $T_a$ and other factors varied seasonally and may have been more consistent in the same month. It was confirmed in the research of Yao et al. (2019) that modeling data of the same month together could achieve more accurate results. Therefore, although day of year was used in the modeling in this study, this temporaltemporary difference was not completely eliminated. Modeling the datasets of all seasons together in this study may increase the temporaltemporary heterogeneity of accuracy. It is worthwhile to consider grouping the data of

475

540 **7 Conclusion**

$T_a$ is a key variable in climate and global change research. In this study, we developed an all-sky 1 km daily mean $T_a$ product for 2003–2019 over mainland China mainly based on MODIS and GLDAS data using the RF method. An efficient temporaltemporary gap-filling method was first used to fill MODIS LST gaps under cloudy-sky conditions. We predicted $T_a$

**Point 7:** are the station Ta measurements used in the prediction of Ta?

**Response 7:** Thank you for your comments. In this study, the station $T_a$ measurements were not used in the prediction of $T_a$, but were used in model training. The data pairs used for model training and validation consist of input features and station measured $T_a$ at the stations. The input features of the models are LSTs, DSR, ALB, LAI, elevation, GLDAS $T_a$, day of year, latitude, and longitude. And the output variable is daily mean $T_a$.

**Reference:**
Gong, P., Wang, J., Yu, L., Zhao, Y., Zhao, Y., Liang, L., Niu, Z., Huang, X., Fu, H., and Liu, S.: Finer resolution observation and monitoring of global land cover: First mapping results with Landsat TM and ETM+ data, Int. J. Remote Sens., 34, 2607-2654, https://doi.org/10.1080/01431161.2012.748992, 2013.

---

## Author Comment (AC3)

**Response to Referee #2 Comments**

We thank Referee #2 for the valuable and constructive comments on our manuscript. A point-by-point response to all comments is listed below.

**Point 1:** My biggest concern is the statistic results may not be credible over southwest China. In southwest China, it rains in the most time of a year. The sunshine is rare that it is said "sunny weather seldom lasts for more than three days". In other words, the percentage of missing MODIS LST data is over 60% due to the presence of clouds. To control the uncertainty introduced by LST gap-filling, a temporal window of 2 days in this study was used to fill the gaps. Please provide detailed availability of LST for cloudy-sky model using this simple gap-filling method.

**Response 1:** Thank you for your comments. In this study, a simple multi-temporal method was used to fill the MODIS LST gaps. In order to balance the MODIS LST gap-filling rate with the large uncertainty caused by the large time threshold, we have conducted experiments with different time thresholds, and finally decided to set the time threshold of ± 2 days. The ratios of available values of four MODIS LSTs at all stations were 33.2 %, 37.6 %, 32.1 %, and 38.0 %, respectively, which increased to 73.0 %, 77.7 %, 72.4 %, and 77.3 %, respectively, after gap-filling.

Moreover, we counted the validation statistics of 485 stations located in southwest China, and the overall $R^2$ and RMSE were 0.978 and 1.428 K, respectively. The density scatter plot of the estimated $T_a$ against the station observed $T_a$ in southwest China under all weather conditions is shown in Fig. 1. The RMSE histogram of stations in southwest China is shown in Fig. 2, with a mean RMSE of 1.405 K. Of the 485 stations, 315 stations had RMSE values of less than 1.5 K, while only 9 stations had RMSE values of more than 2 K. Therefore, stations in southwest China have generally shown satisfactory performance, we consider this gap-filling method feasible for this study.

[Figure]

Figure 1. Density scatter plot of the estimated $T_a$ against the station observed $T_a$ in southwest China under all weather conditions.

[Figure]

Figure 2. The RMSE histogram of stations in southwest China.

**Point 2:** In section 3.2, GLDAS assimilated Ta was used in three models as input features. In the feature importance of those three models, assimilated Ta ranked first for two cloudy models and was second to Terra nighttime LST for clear-sky models. The second biggest concern for this study is that it seems like GLDAS assimilated Ta determines the RMSE and $R^2$. From the fourth paragraph in introduction part, no author used assimilated Ta as the predictor. Instead, shen [1] only used the soil moisture content, albedo and soil evaporation from GLDAS as predictors. If the ground-based Ta ingested by GLDAS was introduced as the preditor, whether it is a circular reasoning that reach better results? I would suggest removing the assimilated Ta as the predictor for three models.

**Response 2:** Thank the reviewer for making the valuable comments. Since the GLDAS assimilated $T_a$ has well captured the spatial and temporal variation of the actual $T_a$, it is not surprising to see the great contributions of the GLDAS $T_a$. However, GLDAS $T_a$ does not completely determine the RMSE and $R^2$ of our models because many additional inputs have greatly improved the $T_a$ prediction. As shown in Fig. 3, the RMSE values of the GLDAS $T_a$ under three weather conditions are 2.705 K, 2.545 K, and 2.588 K, respectively, while our final models have much better results (RMSE values are 1.342 K, 1.440 K, and 1.396 K, respectively).

[Figure]

[Figure]

Figure 3. Density scatter plots of the estimated $T_a$ and GLDAS assimilated $T_a$ against the station observed $T_a$. (a, c, e) are the RF $T_a$ under three weather conditions, (b, d, f) are the GLDAS assimilated $T_a$ under three weather conditions.

Before conducting this study, we did read the paper of Shen et al. (2020) carefully and conducted some experiments. We believe, also based on our initial experiments, that use of GLDAS $T_a$ as a predictor is a much better choice than GLDAS soil moisture (SM), albedo and evaporation because GLDAS assimilated a huge amount of $T_a$ observations into the model to "control" the calculated $T_a$, while SM, albedo and evaporation are calculated outputs and have much larger uncertainties. The predictors need to be as accurate as possible.

Incorporating GLDAS $T_a$ as our model predictor is not a circular reasoning issue since GLDAS $T_a$ can be considered to be a priori knowledge. Use of a priori knowledge has been the common practice in quantitative remote sensing (Liang, 2004; Liang and Wang, 2019).

In fact, after removing GLDAS $T_a$ as the predictor, the validation statistics of the three models are worsened as shown in Fig. 4, especially for cloudy-sky model II, which does not include MODIS LST at all. RMSE values of the three models were 1.498 K,

1.859 K, and 2.359 K, respectively, which increased by 0.156 K, 0.419 K, and 0.963 K compared with that before removing GLDAS $T_a$, respectively. It proved that GLDAS $T_a$ was used as a priori knowledge in this study, rather than completely determining the prediction results. Therefore, we still keep GLDAS $T_a$ as the predictor.

[Figure]

Figure 4. Density scatter plots of the $T_a$ estimated by the models with GLDAS $T_a$ removed against the station observed $T_a$.

**Point 3:** The smallest concern is the spatiotemporal model validation strategy in this study which just relys on random cross validation.
However, ignoring spatial and time dependence in model cross-validation can create false confidence in model predictions and hide model overfitting, and this problem that has been well documented in recent works [2, 3]. Please give explanations why this study still used an overoptimistic approach (random cross validation) to assess the prediction error in both space and time.

**Response 3:** Thank you for your nice comments. To test the models' performance in predicting conditions beyond the temporal and spatial location of the training data, we further used the two validation strategies of Leave-Time-Out (LTO) cross-validation

(CV) and Leave-Location-Out (LLO) CV on the basis of random sample validation. These two strategies have been used in some studies to evaluate the performance of spatiotemporal models in unknown time or unknown space (Liu et al., 2020; Ploton et al., 2020; Xiao et al., 2018).

First, for LTO CV, we divided the data pairs from 2003 to 2016 into 14 groups by calendar year. In each iteration, 13 groups of data were used as training set for model training, and the remaining one group of data was used for validation. The modeling and validation process were repeated 14 times until each year's data was validated. The results are shown in Fig. 5. The RMSE values of validation results for different groups of data range from 1.359 K to 1.665 K. The minor difference between the LTO CV results proves that these models have good extensibility in time.

[Figure]

Figure 5. Density scatter plots of LTO CV results for three models.

Then, for LLO CV, we divided 7 clusters in the Chinese region as shown in Fig. 6 by using the similar separation strategy of Xiao et al. (2018). Stations used in this study were divided into different clusters according to their spatial locations, and all data pairs were divided into 7 groups according to the cluster of station. In each iteration, 6 groups of data were used as training set and the remaining one group of data was used for

validation. The modeling and validation process were repeated 7 times until the data of each group was validated. The total validation results of the models under three weather conditions are shown in Fig. 7, with RMSE values ranging from 1.615 K to 1.957 K. As expected, the prediction error of LLO CV increased relative to random sample validation. This is because the relationship between $T_a$ and other features varies with geographical location. The prediction error of the Northwest and Southwest clusters was larger than that of other clusters. RMSE values of these two clusters exceeded 2.5 K under cloudy-sky conditions II while RMSE values of the other clusters were about 1.5 K. This is consistent with the analysis of the spatial distribution of model accuracy in section 4.4 of the manuscript. The meteorological stations in Northwest China and the Qinghai-Tibet are distributed discretely and far away from other stations in China, leading to a large difference between the training set and the test set, and ultimately resulting in the relatively poor performance in the LLO CV strategy in these two regions. Furthermore, the LLO CV results of the cloudy-sky model II are worse than those of the clear-sky model and cloudy-sky model I, indicating that LSTs help to reduce the spatial overfitting of the models.

We have added the content on page 13-14, lines 275-284 and page 19-21, lines 370-397 in the revised manuscript:

[Figure]

Figure 6. Cluster separation in the research area. According to geographical distribution, mainland China is divided into 7 clusters, which are the North, the Northeast, the Northwest, the Southeast, the relatively cold north, the Qinghai-Tibet Plateau, and the Pearl River Delta, respectively

[Figure]

Figure 7. Density scatter plots of LLO CV results for three models.

[revised manuscript text omitted]

---

## Author Response (AR2)

**Response to Editor's Comments**

We thank Dr. Carlson for the advice on our manuscript. Our response is listed below.

**Point 1:** One final change. Following Copernicus policy for all journals, and having seen this issue arise several times. ESSD asks that authors remove South China Sea inset on all maps (e.g. Figures 1, 11, 14) unless data from that region proves useful or definitive in the manuscript. In this case you do not even show data from that oceanic region and it hardly fits your use of the term 'Mainland China' in your title. Maps in that format represent political provocation not appropriate in science journals; colleagues, editors and reviewers all object. Other authors have made necessary changes; please do likewise.

**Response 1:** Thank you for your advice. We have made the corresponding changes in the relevant figures (Figures 1, 11, 14). The changes can be seen in the updated edition of the manuscript.